# MixMax: Distributional Robustness in Function Space via Optimal Data Mixtures

**Anvith Thudi**
Department of Computer Science
University of Toronto and Vector Institute
anvith.thudi@mail.utoronto.ca

**Chris J. Maddison**
Department of Computer Science
University of Toronto and Vector Institute
cmaddis@cs.toronto.edu

## Abstract

Machine learning models are often required to perform well across several pre-defined settings, such as a set of user groups. Worst-case performance is a common metric to capture this requirement, and is the objective of group distributionally robust optimization (group DRO). Unfortunately, these methods struggle when the loss is non-convex in the parameters, or the model class is non-parametric. Here, we make a classical move to address this: we reparameterize group DRO from parameter space to function space, which results in a number of advantages. First, we show that group DRO over the space of bounded functions admits a minimax theorem. Second, for cross-entropy and mean squared error, we show that the minimax optimal mixture distribution is the solution of a simple convex optimization problem. Thus, provided one is working with a model class of universal function approximators, group DRO can be solved by a convex optimization problem followed by a classical risk minimization problem. We call our method MixMax. In our experiments, we found that MixMax matched or outperformed the standard group DRO baselines, and in particular, MixMax improved the performance of XGBoost over the only baseline, data balancing, for variations of the ACSIncome and CelebA annotations datasets.

## 1 Introduction

Machine learning pipelines often optimize for a model that performs well over several different distributions of data. This can be as the model will be deployed for different groups of users (Strack et al., 2014; Blackard, 1998), or in the development of foundation models for a suite of tasks. Optimizing for the worst-case error over the set of distributions captures this distributional robustness objective, and is called distributionally robust optimization (DRO) (Dupačová, 1987; Shapiro and Kleywegt, 2002; Rahimian and Mehrotra, 2019; Oren et al., 2019; Sagawa et al., 2019). This is often called group DRO (Sagawa et al., 2019) when the distribution set is finite.

Despite the importance of obtaining group DRO models, we lack effective methods for modern machine learning models. When our loss is convex in our model parameters, methods exist for solving the group DRO optimization (Shapiro and Kleywegt, 2002). But in the case of modern expressive non-linear models, only heuristic methods exist (such as applying methods from the convex literature) (Xie et al., 2024; Sagawa et al., 2019). Furthermore, these methods are expensive, unstable, and training on simple balanced mixtures of the data distributions sometimes leads to better group DRO solutions (Idrissi et al., 2022). Moreover, no group DRO methods (beyond balancing data) exist for non-parametric models. We turn to optimizing data mixtures as a possible solution.

Intuitively, mixing the distributions we want robustness over plays an important role in obtaining distributionally robust models. Past work has already shown that how one chooses to weigh sources of data in the loss function has a significant impact on the performance of the final model (Sorscher et al., 2022; Xie et al., 2024) across tasks, a key consideration for large language model training, *e.g.*, Section 3.3 of Bi et al. (2024) and the survey by Albalak et al. (2024). Other work has also already shown that in many cases, DRO solutions minimize a specific mixture (Arjovsky et al., 2019; Słowik and Bottou, 2021; 2022) (existentially). Yet, it is not known what the best data mixture for group DRO is. Minimax theorems provide one way of answering this, as they state a specific

mixture distribution whose optima solve the DRO objective. However, as the loss functions over the parameters of commonly used model classes (like neural networks) are non-convex, convex minimax theorems do not apply. We address this with a classic trick: working over function space.

Modern model classes are able to approximate any function arbitrarily well. Hence, when viewed in function space, these model classes cover all the bounded functions. This subset is also the hypothesis space covering all the Bayes optimal functions (for classification and bounded regression). So we can expect that the group DRO solution is within this subset for modern model classes. Hence, in this paper we study the group DRO problem over the set of bounded functions.

Highlighting the benefits of this, we show that the space of bounded functions admits a minimax theorem. That is, if we can fit any distribution to Bayes optimal, then fitting to a mixture with the highest Bayes error solves the group DRO problem. To obtain this minimax result we bypassed issues with $L^p$ metrics by using topological dual spaces, which may be of independent interest.

Towards now finding this group DRO solving mixture, we show that these optimal mixture distributions *maximize a concave objective function* over the mixture weights. This result is specific to using cross-entropy and $\ell_2^2$ losses, and leverages the structure of their Bayes optimal functions. We further show that one can compute the gradients for this objective given the optimal predictor for each component distribution. Hence, with this gradient oracle, we can use a stochastic entropic mirror ascent algorithm to maximize the objective. We call this selection of mixture weights *MixMax*. In practice we do not have the optimal predictor for the gradient oracle, but we show experimentally several empirical gradient estimators still lead to near optimal mixture weights.

Beyond the guarantees when fitting near optimal, MixMax has a number of advantages for group DRO. First, given a sufficiently large set of data from each component distribution, finding MixMax weights can be accomplished by fitting a separate model on each source—the same amount of training compute as training one model on all the data. Furthermore, because the weights can be used to ensemble the component models, there is little additional model training overhead. Moreover, unlike previous methods for group DRO, MixMax can be used with non-parametric model classes, like gradient boosting (Friedman, 2001) [1].

To illustrate the empirical performance of MixMax when we cannot ensure optimal fitting, we applied it for two real-world model classes. First, we tested MixMax with transformer models on synthetic Markov chain data. Second, we tested MixMax with XGBoost (Chen and Guestrin, 2016) on several tabular datasets with different group shifts (Ding et al., 2021; Liu et al., 2015). In all cases, we found that empirical versions of MixMax matched or outperformed applicable baseline methods when improvement was possible. In particular, when a moderate label shift was present, MixMax yielded relative test accuracy improvements between $2.3 - 5.9\%$ for XGBoost on variations of ACSIncome (Ding et al., 2021) and CelebA annotations (Liu et al., 2015). Our contributions are:

1. A minimax theorem for DRO over bounded functions.
2. Showing that applied to cross-entropy and $\ell_2^2$, this yields a concave objective to maximize for data mixing (to solve group DRO) which we call MixMax
3. Experiments showing empirical versions of MixMax improved over group DRO alternatives
4. Providing the first group DRO method applicable to non-parametric learning, and applying it to XGBoost to improve over the baseline of balancing data by upsampling.

## 2 PRELIMINARIES

In this paper we will consider solving group DRO when we have model classes that are sufficiently expressive to capture the Bayes optimal function for any distribution we are dealing with. This will be relaxed in Section 4.2, but for now this formally means we can (and will) reparameterize our optimization to be over a generic function space.

We work with functions $f : \mathcal{X} \to \mathcal{O}$ from an input domain $\mathcal{X}$ (with a measure $dx$) to a closed convex output domain $\mathcal{O} \subset \mathbb{R}^n$ that have bounded $L^\infty$ norm in each output coordinate: we denote this

---

[1]To the best of our knowledge, there exists no past work on group DRO for non-parametric learning algorithms, with the only work on DRO in general being for k-nearest neighbours where the set of distributions forms a Wasserstein ball (Chen and Paschalidis, 2019).

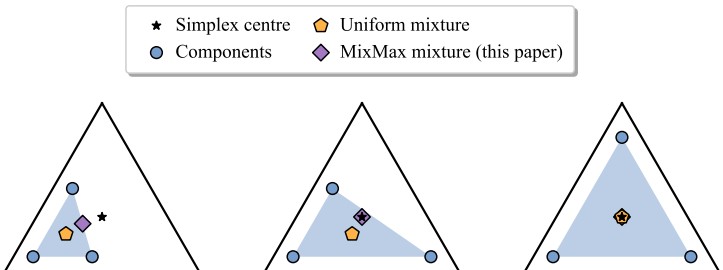

Figure 1: MixMax for classification picks the label probability that maximizes entropy (is closest to the centre of the simplex) in the convex hull of the distributions. We illustrate the label probabilities given by MixMax compared to balancing the distributions when there is only one input and the objective is to minimize worst-case cross-entropy loss.

function space by $L^\infty(\mathcal{X}, \mathcal{O})$. For example, when doing classification $\mathcal{O}$ could be the probability simplex, with the i'th entry representing the probability of label $i$. We will specifically consider the set of functions with outputs bounded by some fixed value $r$ in every coordinate (a.e.), denoted by $B_{L^\infty(\mathcal{X}, \mathcal{O})}(r)$. We work with data $(x, y)$ where $x \in \mathcal{X}$ and $y \in \mathcal{Y}$, and an associated loss function $\mathcal{L}(o, y) : \mathcal{O} \times \mathcal{Y} \to \mathbb{R}^+$ that is convex in the first argument and continuous in both. An example is cross-entropy where $o$ is the label probabilities, and $y$ is a specific label. With $\mathcal{L}$ we have the expected loss of a function $f \in L^\infty(\mathcal{X}, \mathcal{Y})$ over a distribution $dp$ on $\mathcal{X} \times \mathcal{Y}$ is $\int_{\mathcal{X} \times \mathcal{Y}} \mathcal{L}(f(x), y) dp(x, y)$. Given a set of distributions $P$, the DRO objective we study is (which is group DRO when $P$ is finite)

$$\inf_{f \in B_{L^\infty}(r)} \sup_{dp \in P} \int \mathcal{L}(f(x), y) dp(x, y). \tag{DRO}$$

## 3 DRO Over Bounded Functions is Solved by Data Mixing

We now show that, letting our model class be all bounded functions, fitting to an optimal data mixture returns a function that solves group DRO (Corollary 3.2). This is done by observing the set of all bounded functions has a sufficient amount of regularity under the right topology (i.e., metric). We further show that this optimal mixture is characterized by having the highest Bayes error. Later in Section 4 we demonstrate how to optimize for this mixture in the case of cross-entropy and $\ell_2^2$ loss.

First, Theorem 3.1 formally states that there exists a minimizer to the hardest distribution which solves DRO. To show this we overcome challenges in applying Sion's minimax theorem to bounded functions, in particular, the fact that the set is not compact in any $L^p$ topology. Our main contribution is to show that working with the weak*-topology resolves these issues. This follows from leveraging Banach-Alaoglu's theorem alongside the method of test functions, and introduces mild additional assumptions. For example, we will require that $dx$ is a $\sigma$-finite measure to apply $L^p$ duality theory, which is satisfied if $dx$ is Lebesque measure on $R^n$ or counting measure on some countable set. We will further require that $\mathcal{L}$ is bounded on $\mathcal{O} \times \mathcal{Y}$ to use intergral convergence theorems, and discuss the strength of this assumption later. A complete proof is provided in Appendix A.1

**Theorem 3.1** (DRO over $L^\infty$ = DM). *Let $P$ be a set of probability distributions $dp$ on the product space $\mathcal{X} \times \mathcal{Y}$ with $\mathcal{Y} \subset \mathbb{R}^n$ for some $n$, such that $\forall dp \in P$, $dp(x)$ is absolutely continuous w.r.t a given $\sigma$-finite measure $dx$ on $\mathcal{X}$. Let $\mathcal{O} \subset \mathbb{R}^m$ be a closed convex set, and $L^\infty(\mathcal{X}, \mathcal{O})$ be defined w.r.t the measure $dx$, and $B_{L^\infty(\mathcal{X}, \mathcal{O})}(r) = \{f \in L^\infty(\mathcal{X}, \mathcal{O}) : ||f||_\infty \le r\}$.*

*Let the loss function $\mathcal{L}(o, y)$ be continuous in both arguments, and convex in the first argument. Furthermore assume $\mathcal{L}$ is bounded by some constant $M$ on $\mathcal{O} \times \mathcal{Y}$. If $dp_\lambda$ realizes*

$$\sup_{dp \in Conv(P)} \inf_{f \in B_{L^\infty(\mathcal{X}, \mathcal{O})}(r)} \int \mathcal{L}(f(x), y) dp(x, y),$$

*and there exists an $f^* \in B_{L^\infty(\mathcal{X},\mathcal{O})}(r)$ realizing the DRO objective, then there exists a minimizer $f_\lambda$ of the expected loss under $dp_\lambda$ that also realizes the DRO objective*

$$\inf_{f \in B_{L^\infty(\mathcal{X},\mathcal{O})}(r)} \sup_{dp \in P} \int \mathcal{L}(f(x),y)dp(x,y).$$

*That is, the DRO objective is solved by fitting a specific distribution in the convex hull of $P$. If the set of distributions $P$ is finite, then the DRO objective is solved by fitting a specific mixture distribution.*

While bounded loss may seem strong, it can be enforced by choosing the output space $\mathcal{O}$ carefully, e.g., for cross-entropy loss if we know the minimum probability of any label is $\epsilon > 0$ we can choose $\mathcal{O}$ accordingly and avoid loss blow-up (from $log(0)$). Note, our actual requirement was a condition to enforce that pointwise convergence in the loss at every $x$ implies convergence of the integral for all $dp \in P$, which can be true without loss boundedness depending on $P$. On assuming a DRO solution over $B_{L^\infty(\mathcal{X},\mathcal{O})}(r)$ exists, this is true for finite $P$ (Appendix A.1) and hence applies for group DRO.

For the rest of the paper, We will assume there is sufficient regularity (whether imposed by $\mathcal{O}$, $P$, or both) for pointwise convergence of loss to imply the average loss converges and hence the result of Theorem 3.1 applies. We will further assume that for some $r < \infty$ we have the bounded functions $B_{L^\infty(\mathcal{X},\mathcal{O})}(r)$ contains the Bayes optimal solutions for all $dp \in P$ (to leverage structural properties of cross-entropy and $\ell_2^2$). Note the last two assumptions are satisfied for cross-entropy if a.e. all labels have $\geq \epsilon$ probability for some $\epsilon > 0$ (and we choose $\mathcal{O}$ accordingly), and for $\ell_2^2$ if the allowed $y$ values are bounded. We leave it to future work to consider applying Theorem 3.1 without these assumptions, or generalizing Theorem 3.1 itself.

### 3.1 GROUP DRO BY MAXIMIZATION

We now consider applying Theorem 3.1 for group DRO. For cross-entropy and $\ell_2^2$ losses, the Bayes optimal functions are unique up to a measure 0 set over $\mathcal{X}$, and so any minimizer of the hardest distribution is DRO optimal by Theorem 3.1 (given a DRO solution exists which is true for group DRO): all minimizers have the same average losses on $dp \in P$. In this case, if the set $P$ is also finite (i.e., group DRO) we further have the max-min optimization over bounded functions reduces to single maximization over a finite dimensional simplex $\Delta^P$.

**Corollary 3.2** (Group DRO by Maximization). *Let $P$ be a finite set, and assume $\int \mathcal{L}(f(x),y)dp(x,y)$ is uniquely minimized over functions in $B_{L^\infty(\mathcal{X},\mathcal{O})}(r)$ (upto a measure 0 set). Parameterize $Conv(P)$ by $\lambda \in \Delta^P$ via $Conv(P) = \{dp_\lambda := \sum_{dp \in P} \lambda_p dp : \lambda \in \Delta^P\}$. Further denote the minimizer of $\int \mathcal{L}(f(x),y)dp_\lambda(x,y)$ in $B_{L^\infty(\mathcal{X},\mathcal{O})}(r)$ by $f_\lambda$ (parameterized by $\lambda$). Then for $\lambda$ realizing $\sup_{\lambda \in \Delta^P} \int \mathcal{L}(f_\lambda(x),y)dp_\lambda(x,y)$, $f_\lambda(x)$ realizes $\inf_{f \in B_{L^\infty(\mathcal{X},\mathcal{O})}(r)} \sup_{dp \in P} \int \mathcal{L}(f(x),y)dp(x,y)$. That is, the group DRO optimization over $L^\infty$ reduces to a single maximization over $\Delta^P$.*

*Proof.* Follows from Theorem 3.1, noting $P$ is finite so a DRO solution exists, and rewriting the condition for the mixture with the parameterizations given by uniqueness of solutions. $\square$

## 4 MIXMAX

We now apply Theorem 3.1 (under the assumptions of the previous section) to cross-entropy and $\ell_2^2$ for finite $P$. We use $p(x)$ and $f_p(x)$ to represent the covariate density and Bayes optimal function of distributions $dp \in P$, and $p_\lambda(x)$ and $f_\lambda(x)$ similarly for the mixture distributions $dp_\lambda \in Conv(P)$.

First note that, by Corollary 3.2, the objective for the optimal mixture weights is

$$\sup_{\lambda \in \Delta^P} \int \mathcal{L}(f_\lambda(x),y)dp_\lambda(x,y) = \sup_{\lambda \in \Delta^P} \sum_{dp \in P} \lambda_p \int \mathcal{L}(f_\lambda(x),y)dp(x,y) \tag{1}$$

We show that this is concave and that we can compute its gradients for cross-entropy and $\ell_2^2$. Thus, we can perform entropic mirror ascent Duchi (2018) to solve the constrained optimization. Algorithm 1 describes this approach, which we call MixMax ("Mixtures by Maximization").

---

**Algorithm 1** Empirical MixMax

---

**Require:** Step size $\eta$, number of steps $n$, loss function $\mathcal{L}$ (either cross-entropy or $\ell_2^2$), and, for each distribution $dp$ in the set $P$, samples $D_p$, proxy/exact covariate density $p(x)$, and proxy/Bayes optimal prediction function $f_p$.

**Note:** If there is no covariate shift then one can set $p(x) = q(x) \; \forall dp \in P$ for any fixed $q(x)$; this has no impact due to symmetry in the formula used in the algorithm.

**Initialize:** $\lambda_p \leftarrow \frac{1}{|P|}$ for all $dp \in P$

1: **for** $i = 1, \ldots, n$ **do**
2:     $f_\lambda(x) \leftarrow \frac{\sum_{p \in P} \lambda_p p(x) f_p(x)}{\sum_{p \in P} \lambda_p p(x)}$
3:     $l \leftarrow \sum_{dp \in P} \frac{\lambda_p}{|D_p|} \sum_{(x,y) \in D_p} \mathcal{L}(f_\lambda(x), y)$
4:     $g \leftarrow \nabla_\lambda l$
5:     $\lambda_p \leftarrow \frac{\lambda_p e^{\eta g_p}}{\sum_{dp \in P} \lambda_p e^{\eta g_p}}$ for all $dp \in P$
6: **end for**
7: **return** $\{\lambda_p\}_{dp \in P}$

---

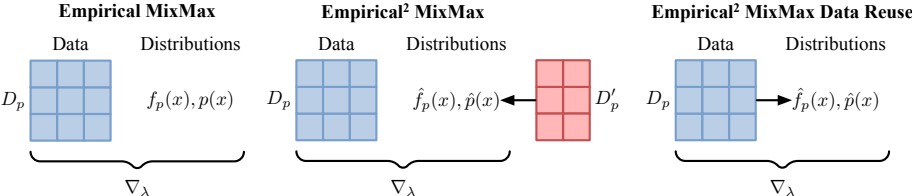

Figure 2: An illustration of the data requirements for the empirical MixMax gradients.

**The Objective is Concave** In the case of cross-entropy, this objective reduces to the expected entropy of $y$ conditioned on $x$, $p_\lambda(y|x)$, over $x \sim p_\lambda(x)$. Therefore it is a concave maximization problem in the mixture weights via the concavity of entropy. In the case of $\ell_2^2$, under our assumptions, this objective reduces to the expectation of the conditional variance of $y$ given $x$ over $x \sim p_\lambda(x)$, which is again a concave objective. Appendix A.2 provides full proofs of concavity.

## 4.1 Computing Gradients of the MixMax Objective

We now demonstrate how to compute the gradient of the objective in Equation 1, i.e., $\sum_{dp \in P} \int \nabla_\lambda \lambda_p \mathcal{L}(f_\lambda(x), y) dp(x, y)$ [2], for cross-entropy and $\ell_2^2$ given $f_p$ and $p(x)$ and the ability to integrate for all $dp \in P$. We present this in cases, and later discuss empirical implementations.

**Case 1: No Covariate Shift** Suppose there is no covariate shift, *i.e.*, $\exists \; p_0(x)$ such that $p(x) = p_0(x) \; \forall dp \in P$. Then, for cross-entropy we have the Bayes optimal function $f_\lambda(x) = p_\lambda(y|x) = \sum_{dp \in P} \lambda_p p(y|x) = \sum_{dp \in P} \lambda_p f_p(x)$ and for $\ell_2^2$ we also have $f_\lambda(x) = \mathbb{E}_{y \sim p_\lambda(y|x)} y = \sum_{dp \in P} \lambda_p \mathbb{E}_{y \sim p(y|x)} y = \sum_{dp \in P} \lambda_p f_p(x)$. Hence given $f_p(x) \; \forall dp \in P$ we can compute $f_\lambda(x)$ and $\nabla_\lambda f_\lambda(x)$. Hence, $\nabla_\lambda \lambda_p \mathcal{L}(f_\lambda(x), y)$ follows by product and chain rule, and given the ability to integrate over $dp \in P$ we can compute the gradient of the objective.

**Case 2: Covariate Shift** The more general expression under covariate shift, for cross-entropy and $\ell_2^2$, is $f_\lambda(x) = \frac{\sum_{dp \in P} \lambda_p f_p(x) p(x)}{\sum_{dp' \in P} \lambda_{p'} p'(x)}$. Hence computing $f_\lambda(x)$ and $\nabla_\lambda f_\lambda(x)$ would further require knowledge of $p(x) \; \forall dp \in P$. Given this, computing $\nabla_\lambda \lambda_p \mathcal{L}(f_\lambda(x), y)$ follows by product and chain rule, and given the ability to integrate over $dp \in P$ we can compute the gradient of the objective.

## 4.2 EMPIRICAL MIXMAX GRADIENTS AND PRACTICAL CONSIDERATIONS

Section 4.1 described how to compute the gradients of the MixMax objective given we have the Bayes optimal function (and covariate density function if necessary) for each distribution, and can integrate exactly over the domain. We now decribe empirical gradients which remove these assumptions, summarized in Figure 2. These add varying sources of empirical error.

**Empirical MixMax (EMixMax)**   Computing the MixMax gradients is often intractable due to the integral over $dp \in P$. However, given datasets $D_p$ for $dp \in P$ one can compute empirical MixMax (EMixMax) gradients and update $\lambda$ with stochastic entropic mirror ascent (Algorithm 1):

$$\nabla^{\text{EMixMax}}(\lambda) = \sum_{dp \in P} \frac{1}{|D_p|} \sum_{(x,y) \in D_p} \nabla_\lambda \lambda_p \mathcal{L}(f_\lambda(x), y).$$

We prove that this is an unbiased estimator and discuss its computational complexity in Appendix B.

**Empirical² MixMax**   One also usually does not know the Bayes optimal functions $f_p(x)$ or the densities $p(x)$. However one can train a model on samples from $dp \in P$ to obtain approximations $\hat{f}_p(x)$ and $\hat{p}$, and then use another set of samples to compute $\nabla^{EMixMax}$. We call this approach Empirical² MixMax or E²MixMax. More precisely, assuming no covariate shift, given i.i.d datasets $D_p, D'_p \ \forall dp \in P$, the E²MixMax gradients are

$$\nabla^{\text{E}^2\text{MixMax}}(\lambda) = \sum_{dp \in P} \frac{1}{|D_p|} \sum_{(x,y) \in D_p} \nabla_\lambda \lambda_p \mathcal{L}\left(\sum_{dp \in P} \lambda_p \hat{f}_p(x), y\right)$$

$$\text{where } \hat{f}_p(x) = \arg\min_f \frac{1}{|D'_p|} \sum_{(x,y) \in D'_p} \mathcal{L}\left(f(x), y\right).$$

It is challenging to evaluate the expectation of $\nabla^{\text{E}^2\text{MixMax}}$, and we leave this to future work. Instead, we study this estimator empirically and find it gives results close to EMixMax. In the case of covariate shift, we also fit density approximations $\hat{p}(x)$ on the second dataset $D'_p$ and use the covariate shift definition of $f_\lambda$. Algorithm 1 describes the changes for E²MixMax . We also consider *E²MixMax with Data Reuse*, i.e., $D_p = D'_p$, as a more sample efficient alternative.

**How to Use MixMax Weights**   Finally, MixMax mixture weights can be used in two ways. We can use the weights to define a mixture distribution and fit a new model on it, which Theorem 3.1 states will result in a group DRO solution if we fit optimally. Alternatively, we can use the mixture weights to combine our (approximate) $f_p$ models (i.e., the $f_\lambda$ formula), which also returns an approximation of the best model for the MixMax mixture distribution. These approaches provide a trade-off: the first approach has higher training cost but results in a single model and hence smaller inference cost. We employed both approaches in our experiments and found they both improved over baselines.

## 4.3 ILLUSTRATING MIXMAX

Here we investigated patterns for the group DRO solutions found by EMixMax, running it with many samples as to be representative of the true group DRO solution. To understand Binary Classification we considered two cases: one where the two distribution were mirror opposites of each other, and one where this was not the case. Specifically we considered binary classification distributions with the same covariate probabilities $p(x) = unif[0, 1]$, but with $p_1(1|x) = 0.5cos(\pi x) + 0.5$ and $p_2(1|x) = -0.5cos(\pi x) + 0.5$ (Figure 3a) or $p_2(1|x) = -0.5cos(\pi(x - 0.2)) + 0.5$ (Figure 3b). Note the MixMax objective maximizes average entropy within the mixture span, and in the first case we found EMixMax selected random guessing which is the maximal entropy distribution

---
[2]Formally, we require $\nabla_\lambda \mathcal{L}(f_\lambda(x), y)$ to have sufficient regularity (e.g., is bounded).

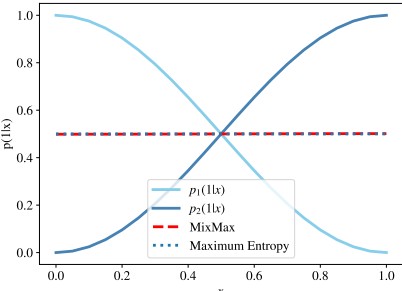 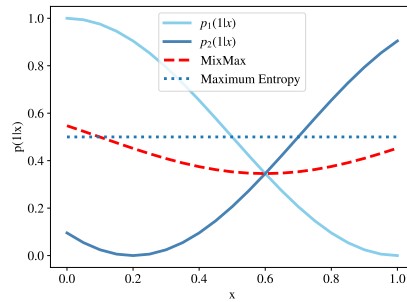

(a) Maximum Entropy Conditional in Mixture Span   (b) Maximum Entropy Conditional **not** in Mixture Span

Figure 3: EMixMax for cross-entropy maximizes average prediction entropy within the mixture span.

(the true group DRO solution). In the second case where random guessing is not in the span, EMixMax found a mixture closer to random guessing than the individual distributions (expected for the true group DRO solution). For regression we considered deterministic predictions and varied which distributions had the extreme predictors. Specifically, we considered a fixed $p(x) = unif[0, 1]$, but with $p_1(1|x) = 0.2cos(\pi x) + 0.5$, $p_2(1|x) = 0.1$, and $p_3(1|x) = 0.15$ or setting $p_3(1|x) = 0.8$ (Figure 6a) and Figure 6b in Appendix D respectively). We found EMixMax always found the extreme functions and balanced those, which minimizes maximal $\ell_2^2$ error (true group DRO solution).

## 5  RELATED WORK

**Distributionally Robust Optimization**   DRO is motivated by several applications not limited to just machine learning, such as resource allocation (Dupačová, 1987). Approaches for DRO often focus on "uncertainty" sets $P$ (the set of distributions we are minimizing the maximum error over) that admit a duality theory (Rahimian and Mehrotra, 2019; Delage and Ye, 2010; Bertsimas et al., 2010; Wiesemann et al., 2013; Ben-Tal et al., 2013), such as Lagragian duality, conic duality, Fenchel duality, etc. For theorem 3.1 we assume nothing on $P$, and for some results, that it is finite similar to work on the Cutting-Surface approaches to DRO (Rahimian and Mehrotra, 2019; Pflug and Wozabal, 2007; Rahimian et al., 2019; Bansal et al., 2018); the finite $P$ case is widely called group DRO (Sagawa et al., 2019) in deep learning for its applications where well-specified "groups" exist in the collected data. In particular, for finite $P$ the equivalence to the convex hull of the set of distributions used in Theorem 3.1 already appears as Lemma 17 in Rahimian and Mehrotra (2019) and is wide-spread in the finite $P$ literature. On similarity in theory, our main theorem that DRO is solved by data mixing is most closely related to the saddle point approaches to stochastic programming (Shapiro and Kleywegt, 2002; Dupačová, 1987; Nemirovski et al., 2009; Zhang et al., 2024; Yu et al., 2024). These results present duality theory where solving for the hardest distribution is enough when the set of parameters to minimize w.r.t is a subset of $\mathbb{R}^n$. In the case of Dupačová (1987) a focus is placed on sets of all distributions meeting certain moment constraints (e.g., only the mean and variance are known). Shapiro and Kleywegt (2002); Zhang et al. (2024); Yu et al. (2024) consider the case of an arbitrary finite set of distributions like our work, proposing sample average approaches for optimal solutions. Our method also uses sample averages, but further leverages the structure of cross-entropy and $\ell_2^2$ loss over bounded functions to collapse the max-min optimization to a single maximization.

Nevertheless, to the best of our knowledge, our work is the first to extend these past minimax approaches to DRO over the set of all bounded functions (Theorem 3.1). This yields a concave objective to maximize (Corollary 3.2) which we find yields good results for practical applications involving non-parametric learning and expressive non-linear model classes.

**Data Mixing**   The work of Xie et al. (2024), building on past work on DRO for deep learning (Sagawa et al., 2019; Oren et al., 2019), highlighted empirically how optimizing dataset mixtures can lead to better performance over several downstream tasks. However, it is not clear whether this is due to faster convergence or due to the optima being better suited for the set of downstream tasks. On this, past work has highlighted the role of dataset selection for convergence rates (Sorscher et al.,

2022; Kolossov et al., 2023), alongside increasing sample access (Jain et al., 2024). In this paper we asked what the role of dataset selection, in particular data mixing, is for having optima better suited for deployments with uncertain downstream tasks. The invariant risk minimization (Arjovsky et al., 2019) and data bias (Słowik and Bottou, 2021; 2022) literature have considered the same question, but the analysis there has been limited to problems satisfying the KKT conditions or to studying local minima in $\mathbb{R}^n$, and do not discuss how to find the best data mixture. We also note recent work by Fan et al. (2023) proposed data mixing methods that build on Xie et al. (2024), but departed away from the DRO objective (and Xie et al. (2024) performed comparably with enough compute).

## 6 EXPERIMENTS

We only proved the guarantees of MixMax when we can obtain the optimal model over all bounded functions for our distributions. This is often not possible in practice. Hence, we tested how empirical implementations of MixMax (described in Section 4.2) performed for real-world model classes with only finite samples from each distribution; we observed MixMax still improved over the baselines even when such empirical errors were introduced.

In our experiments, we ran EMixMax for 10 steps with $\eta = 2.0$ for all the sequence modeling tasks, and for 20 steps with $\eta = 0.1$ for the tabular datasets unless otherwise specified; preliminary testing showed that this was enough to have the objective converge within $0.01$ between iterates. We used Nvidia RTX 2080 Ti and A100 GPUs to accelerate our experiments involving small transformers, and otherwise used Intel Xeon Silver 4210 CPUs and AMD EPYC 7643 CPUs. We used the GPTNeo architecture (Black et al., 2021) for the transformers (hyperparameters described in Appendix C).

### 6.1 EMixMax AND E²MixMax VARIANTS MAXIMIZE MixMax OBJECTIVE COMPARABLY

Here we investigated how E²MixMax with various data splits compared to EMixMax in maximizing the MixMax objective. Specifically, we considered 4 tokens $\{0, 1, 2, 3\}$ and sequences generated by a Markov chain, and the task was to model the mixed distribution of sequences from lengths 1 to 10 where the probability a sequence was of length $i$ was always $1/10$. We constructed three Markov chains to perform group DRO over by independently sampling transition probabilities from a symmetric Dirichlet distribution with magnitude 1.0. We then constructed training datasets by varying the number of samples per length, and considered either using all the samples to perform EMixMax with the ground truth prediction probabilities, E²MixMax with $(75 : 25), (50 : 50)$, and $(25 : 75)$ split between the proxy model (a small transformer) training set and EMixMax set, and E²MixMax with Data Reuse. This is shown in Figure 11 (in Appendix D), where we see that for low samples E²MixMax with Data Reuse performs better than alternative E²MixMax approaches, but as we increase the samples all methods are comparable to EMixMax .

Now just comparing E²MixMax with Data Reuse to E²MixMax with a $(75 : 25)$ split, but varying the similarity in the set of distribution (by changing the Dirichlet magnitude) with a fixed 800 samples per length, we found in Figure 9 (in Appendix D) they perform comparably. We conclude one can effectively resuse training samples to run MixMax and be more sample efficient.

### 6.2 E²MixMax PERFORMS BETTER THAN GROUP DRO BASELINES FOR SEQUENCE MODELING

Here we investigated how E²MixMax compared to other mixture finding methods (balanced and DoReMi (Xie et al., 2024) which follows the near sample optimal algorithm of Zhang et al. (2024) for convex group DRO) and the original gradient descent and ascent group DRO algorithm (Sagawa et al., 2019) across sets of distributions with varying similarity. To do so we considered the same sequence modeling task as Section 6.1, with Markov chain transition probabilities samples from symmetric Dirichlet distributions with magnitudes $1.0, 3.0, 5.0, 7.0, 10.0$ to represent increasing similarity between the Markov chains. For all methods we took a training set of 800 samples per length and a test set of 200 samples per length from each Markov chain. We applied E²MixMax given a small transformer trained for next token prediction on 600 of the 800 training samples per length (leaving the other 200 training samples per length to run EMixMax). We further ran EMixMax with the true probabilities on the 200 held-out samples as a reference for best empirical performance. We applied DoReMi and group DRO using the same small transformer architecture and all 800 training

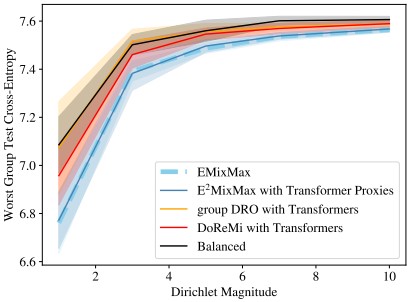
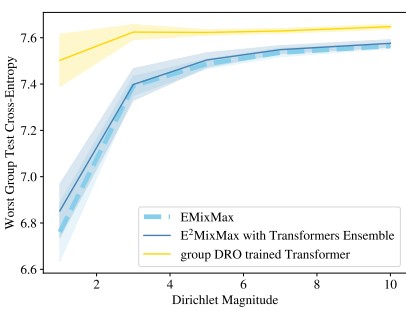

(a) Comparing mixtures found by different methods.

(b) Comparing group DRO Models

Figure 4: E$^2$MixMax found better mixture weights on a sequence modeling task, and its ensemble model performed better than the group DRO trained model. The improvement is stronger when the distributions are less similar. We present the mean and $0.5$ standard deviation of the worst group (i.e., Markov chain) cross-entropy of the Bayes optimal function for the different methods' mixture weights in Figure 4a. Figure 4b further compares the performance between using E$^2$MixMax weights to ensemble its proxy functions and the model given by training with group DRO.

samples per length (per Markov chain). Hyperparameters are described in Appendix C, and we reported the results for each method with the hyperparameter settings that had the lowest maximum group cross-entropy test set loss over $15$ trials of generating sets of Markov chains and samples.

We found E$^2$MixMax returned better mixtures to fit to than other methods, being close to the optimal performance (EMixMax), and that ensembling its proxy transformers with its mixture weights performed better than training with group DRO and was still close to optimal. Specifically, in Figure 4a we compared the worst group cross-entropy loss of the Bayes optimal function for the mixture given by E$^2$MixMax , EMixMax using ground truths to represent optimal performance, DoReMi, the average group DRO mixture weights, and also balanced weights. As seen, fitting to the mixture given by E$^2$MixMax always performed best, doing even better as the distributions became less similar (lower magnitude). We also found the mixture weights returned by group DRO performed comparably to balanced, consistent with previous findings on group DRO performance Idrissi et al. (2022). However, we note that group DRO's intended use is to apply the model it trained and not its mixture weights. In Figure 4b we observed using E$^2$MixMax mixture weights to ensemble its proxy models performed better than the group DRO trained model; note the methods have comparable training compute as the 3 models in the ensemble trained on $1/3$ of the training set each. These results were consistent even if we used fewer training samples (Figures 7b 7a in Appendix D) highlighting that we had sufficient samples for both experiments.

## 6.3 E$^2$MIXMAX WITH DATA REUSE BEATS BALANCING DATA FOR XGBOOST

We compared E$^2$MixMax with Data Reuse weights to data balancing for non-parametric learning algorithms, in particular XGBoost Chen and Guestrin (2016) which is known to be the state of the art for tabular data. We selected ACSIncome Ding et al. (2021) (released under the MIT license) and CelebA annotations Liu et al. (2015) (released for non-commercial use [3]) to test on. For ACSIncome we constructed the dataset from the first 10 American states in alphabetic order, and considered group shifts from race and sex. We further constructed variations of the dataset using all the features, the first 2 features, and the first feature to introduce varying covariate shifts. For CelebA annotations, we used attractiveness as the label with Young and Pale Skin as the group shifts, and constructed variations of the dataset using all features, the first 10 features, and the first 5 features [4]. We used random $80\% - 20\%$ train-test splits in all settings. We applied E$^2$MixMax with Data Reuse by using XGBoost models (trained on the group data) as proxies for label probabilities, and modeled covariate probabilities using Gaussian kernel density estimation. We then returned an XGBoost

---

[3]The agreement for use is at `https://mmlab.ie.cuhk.edu.hk/projects/CelebA.html`.

[4]The different number of features compared to ACSIncome was because the features were binary while ACSIncome first two features take on more values.

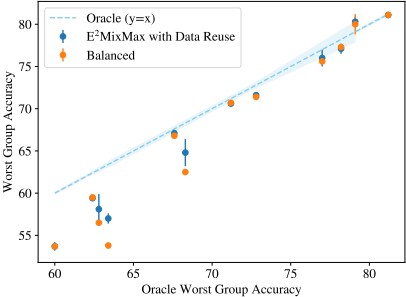 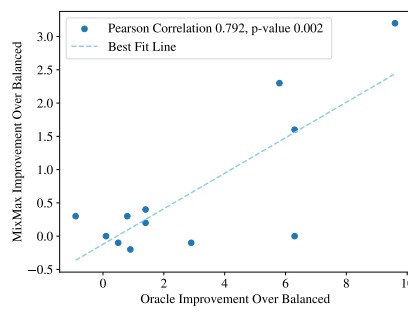

(a) Comparing non-parametric group DRO methods  (b) Correlation with room for improvement

Figure 5: $E^2$MixMax with Data Reuse improved worst group accuracy over balancing data more when there was bigger room for improvement. In Figure 5a we present the mean and $1$ standard deviation (over $5$ trials) of the worst group accuracy of $E^2$MixMax with Data Reuse and balanced data as a function of the oracle accuracy for that setting. In Figure 5b we plot $E^2$MixMax with Data Reuse's improvement in worst group accuracy over balanced data as a function of the Oracle worst group accuracy (i.e., having a model trained for each distribution) improvement over balanced data; we observed a Pearson correlation of $0.792$ with p-value $0.002$.

model trained on the same training data but with the loss reweighed according to weights returned by $E^2$MixMax with Data Reuse. For the balanced baseline, we returned an XGBoost model trained with the data balanced by by normalizing the loss for each group. Lastly, as a measure for best possible performance, we also presented the worst oracle accuracy, where oracle accuracy refers to the test accuracy of an XGBoost model trained on the same distribution (where we need access to the group identity at test time to use this model). Hyperparameters are described in Appendix C, and we reported the results for the hyperparameter setting with the best performance over $5$ trials with random train-test splits.

As seen in Figure 5a ( Table 1 in Appendix D), $E^2$MixMax with Data Reuse matched or outperformed the worst group accuracy of data balancing in all settings, and similarly for worst group loss (Table 3 in Appendix D). In particular, as seen in Figure 5b we observed **the improvement over the worst group accuracy of data balancing was stronger when more room for improvement was possible** (a larger gap between oracle and balanced worst group accuracy): we observed a $0.79$ Pearson correlation factor. In the end, we found $E^2$MixMax with Data Reuse improved worst group accuracy on ACSIncome with groups by sex and one feature by $1.6\%$ ($2.8\%$ relative gain), and CelebA with groups by Young and 10 and 5 features by $2.3\%$ and $3.2\%$ ($3.7\%$ and $5.9\%$ relative gains).

# 7 CONCLUSION

In this paper we showed that group DRO over bounded functions can be solved by fitting to an optimal data mixture, and that maximizing a particular concave objective returns the optimal mixture weights for cross-entropy and $\ell_2^2$ loss. We called this method for finding data mixtures MixMax. Our experiment on a simple sequence modeling task showed that even empirical versions MixMax improved over previous parametric group DRO baselines. An empirical version of MixMax was also shown to improve over the baseline of balancing data for non-parametric learning algorithms, specifically XGBoost, for which no previous group DRO methods were proposed. We leave open the problem of applying our minimax theorem to other losses, and proposing better empirical versions of MixMax.

**Limitations** The empirical versions of MixMax can immediately scale to provide group DRO solutions to large generative modeling over high-dimensional spaces (e.g., decoder-only LLMs) where there is no need to model covariate shifts as there are no inputs. However, a main technical limitation of MixMax methods is the need to model covariate shifts if covariate shifts exist, which often requires large amounts of data in high-dimensional covariate spaces. We also acknowledge that DRO may be used to claim a model is fair, despite it still carrying societal biases. We hope future uses will take care in considering the claims DRO can and cannot make.

ACKNOWLEDGEMENTS

Resources used in preparing this research were provided in part by the Province of Ontario, the Government of Canada through CIFAR, and companies sponsoring the Vector Institute. We acknowledge the support of the Natural Sciences and Engineering Research Council of Canada (NSERC), RGPIN-2021-03445. Anvith Thudi is supported by a Vanier Fellowship from NSERC. We thank Relu Patrascu for administrating and procuring the compute infrastructure used for the experiments in this paper. We would also like to thank Ayoub El Hanchi, Leo Cotta, Nishkrit Desai, Stephan Rabanser, Sierra Wyllie, Alon Albalak, Nicolas Papernot, Colin Raffel, and many others at the Vector Institute for discussions contributing to this paper.

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

## APPENDIX

## A PROOFS

### A.1 PROOF OF THEOREM 3.1

*Proof.* First note $\sup_{dp \in P} = \sup_{dp \in Conv(P)}$ as the objective is linear in $dp$. More formally we have

$$\sup_{dp \in P} \int \mathcal{L}(f(x), y) dp(x, y) \leq \sup_{dp \in Conv(P)} \int \mathcal{L}(f(x), y) dp(x, y)$$

by $P \subset Conv(P)$. Furthermore we have:

$$\sup_{dp \in Conv(P)} \int \mathcal{L}(f(x), y) dp(x, y) = \sup_{\lambda \in \Delta^{|P|}} \int \mathcal{L}(f(x), y) \sum_{dp \in P} \lambda_p dp(x, y)$$

$$= \sup_{\lambda \in \Delta^{|P|}} \sum_{dp \in P} \lambda_p \int \mathcal{L}(f(x), y) dp(x, y)$$

$$\leq \sup_{dp \in P} \int \mathcal{L}(f(x), y) dp(x, y)$$

where we used the parameterization of the convex hull for the first equality, then linearity of the integral for the second, and then the fact the convex combination of a finite set of scalars is less than the maximum scalar in that set. This allows us to conclude $\sup_{dp \in P} \int \mathcal{L}(f(x), y) dp(x, y) = \sup_{dp \in Conv(P)} \int \mathcal{L}(f(x), y) dp(x, y)$. Hence we have, if a minimizer exists for $\sup_{dp \in P} \int \mathcal{L}(f(x), y) dp(x, y)$ then it must also be a minimizer of $\sup_{dp \in Conv(P)} \int \mathcal{L}(f(x), y) dp(x, y)$ (as they are equal for all $f$). Hence we now characterize the minimizer of $\sup_{dp \in Conv(P)} \int \mathcal{L}(f(x), y) dp(x, y)$.

Note we can apply Sion's minimax theorem Sion (1958) to interchange the supremum and infimum if $B_{L^\infty(\mathcal{X}, \mathcal{O})}(r)$ is compact in a topology that the objective is continuous in. Note $B_{L^\infty(\mathcal{X}, \mathbb{R}^m)}(r)$ is compact in the Weak*-topology on $L^\infty(\mathcal{X}, \mathbb{R}^m)$ (as the dual of $L^1(\mathcal{X}, \mathbb{R}^m)$) by Banach-Alaoglu's theorem and continuity of dilation Folland (1999). Now note $B_{L^\infty(\mathcal{X}, \mathcal{O})}(r)$ is a closed subset of $B_{L^\infty(\mathcal{X}, \mathbb{R}^m)}(r)$ and hence is also compact. The fact $B_{L^\infty(\mathcal{X}, \mathcal{O})}(r)$ is closed follows from $\mathcal{O}$ being closed and that weak* convergence implies pointwise convergence (a.e.) in $B_{L^\infty(\mathcal{X}, \mathbb{R}^m)}(r)$ (a set containing $B_{L^\infty(\mathcal{X}, \mathcal{O})}(r)$), which is proven below.

Note convergence in weak* for functions in $B_{L^\infty(\mathcal{X}, \mathbb{R}^m)}(r)$ means the functions converge pointwise a.e. This is as, for $g \in L^1$ and $f$ the limit of $f_n$ in the weak*-topology, $0 = \lim_{n \to \infty} \int (f - f_n) g dx = \int \lim_{n \to \infty} (f - f_n) g dx$ by dominated convergence as functions are bounded. Now taking $g = sign(\lim_{n \to \infty}(f - f_n)) \chi_E$ for some finite measure set $E$, note we have $g \in L^1$ as $\int sign(\lim_{n \to \infty}(f(x) - f_n(x))) \chi_E dx \leq \int \chi_E dx < \infty$. Applying this $g$ to our previous limit expression we have:

$$0 = \int \lim_{n \to \infty} (f - f_n) g dx = \int \lim_{n \to \infty} (f - f_n) sign(\lim_{n \to \infty} (f - f_n)) \chi_E dx$$

$$= \int |\lim_{n \to \infty} (f - f_n)| \chi_E dx$$

By proposition 2.16 in Folland (1999) we have that $\lim_{n \to \infty} (f - f_n)(x) = 0$ a.e on $E$. As the space is countably coverable by finite measured sets (by the $\sigma$-finite assumption), i.e., $\mathcal{X} = \cup_{i=1}^{\infty} E_i$ where $\forall i \int \chi_{E_i} dx < \infty$, this implies the functions converge pointwise a.e in $\mathcal{X}$.

Now as $\mathcal{L}$ is bounded, and the functions converge pointwise in the weak*-topology, we also have by the dominated convergence theorem the objective is continuous in the weak* topology on $L^\infty$. More formally, by continuity of $\mathcal{L}$ in its $\mathcal{O}$ argument, we have if $f_n(x) \to f(x)$ then $\lim_{n \to \infty} \mathcal{L}(f_n(x), y) = \mathcal{L}(f(x), y)$. From this and the previously shown fact that if $f_n \to^{weak^*} f$ then $f_n(x) \to f(x)$ a.e we have for any convergent sequence in the weak*-topology, $f_n \to^{weak^*} f$, that

$$\lim_{n \to \infty} \int \mathcal{L}(f_n(x), y) dp(x, y) = \int \lim_{n \to \infty} \mathcal{L}(f_n(x), y) dp(x, y) = \int \mathcal{L}(f(x), y) dp(x, y)$$

where we pulled the limit through the integral by the boundedness of $\mathcal{L}$ (which allows dominated convergence). Note this is the definition of the objective being continuous in a topology, so we conclude $\int \mathcal{L}(f(x), y) dp(x, y)$ is continuous over the weak*-topology over $f$.

Thus with the weak*-topology on $L^\infty$ we satisfy the compactness and continuity conditions for Sion's minimax theorem, which implies

$$\sup_{dp \in Conv(P)} \inf_{f \in B_{L^\infty(\mathcal{X}, \mathcal{O})}(r)} \int \mathcal{L}(f(x), y) p(x, y) = \inf_{f \in B_{L^\infty(\mathcal{X}, \mathcal{O})}(r)} \sup_{dp \in Conv(P)} \int \mathcal{L}(f(x), y) p(x, y)$$

So if $f^*$ achieves the inf in the LHS and $dp_\lambda$ achieves the sup in the RHS, then note

$$\int \mathcal{L}(f^*(x), y) dp_\lambda(x, y) \leq \sup_{dp \in Conv(P)} \int \mathcal{L}(f^*(x), y) dp(x, y)$$

$$= \inf_{f \in B_{L^\infty(\mathcal{X}, \mathcal{O})}(r)} \int \mathcal{L}(f(x), y) dp_\lambda(x, y)$$

to conclude that $\int \mathcal{L}(f^*(x), y) dp_\lambda(x, y) = \inf_{f \in B_{L^\infty(\mathcal{X}, \mathcal{O})}(r)} \int \mathcal{L}(f(x), y) dp_\lambda(x, y)$. So in particular $f^*$ is a minimizer of $dp_\lambda$. Note minimizers of $dp_\lambda$ exist by the compactness of $B_{L^\infty(\mathcal{X}, \mathbb{R}^m)}(r)$ and continuity of the objective over this function space.

$\square$

**Remark on Existence of DRO Solution**   The theorem required that there exists a DRO solution in the first place, and this is satisfied if $P$ is finite by the continuity of the supremum over a finite set of continuous functions (note continuity of the objective for a single $dp$ was proven above). Further care is needed when $P$ is not finite, but the paper focuses on the finite $P$ case and so we do not discuss this further.

A.2   CONCAVITY OF OBJECTIVES

**Fact A.1.** *If $\mathcal{L}$ is cross-entropy, then $\int \mathcal{L}(f_\lambda(x), y) dp_\lambda(x, y)$ is concave in $\lambda$.*

*Proof.* We note $f_\lambda = p_\lambda(y|x) = \sum_{dp \in P} \lambda_p(x) p(y|x)$ where $\lambda_p(x) = \frac{\lambda_p p(x)}{\sum_{dp' \in P} \lambda_{p'} p'(x)} = \frac{\lambda_p p(x)}{p_\lambda(x)}$. Note that $-\int_{\mathcal{Y}} \log(\sum_{dp \in P} \lambda_p(x) p(y|x)) \sum_{dp \in P} \lambda_p(x) p(y|x) \geq \sum_{dp \in} \lambda_p(x) \int_{\mathcal{Y}} -log(p(y|x)) p(y|x)$ by the concavity of entropy. Applying this we have

$$\int_{\mathcal{X}} \int_{\mathcal{Y}} -log(p_\lambda(y|x))p_\lambda(y|x)p_\lambda(x)$$

$$\geq \int_{\mathcal{X}} \left( \sum_{dp\in P} \lambda_p(x) \int_{\mathcal{Y}} -log(p(y|x))p(y|x) \right) p_\lambda(x)$$

$$= \int_{\mathcal{X}} \left( \sum_{dp\in P} \frac{\lambda_p p(x)}{p_\lambda(x)} \int_{\mathcal{Y}} -log(p(y|x))p(y|x) \right) p_\lambda(x)$$

$$= \sum_{dp\in P} \lambda_p \int_{\mathcal{X}} \int_{\mathcal{Y}} -log(p(y|x))p(y|x)p(x) \quad (2)$$

Note this inequality holds for an arbitrary set of distributions $P$. To now conclude concavity to the mixture weights, let $p_1 = p_{\lambda_1}, p_2 = p_{\lambda_2}$. Then by the above inequality we have $\int \mathcal{L}(f_{\alpha\lambda_1+(1-\alpha)\lambda_2}(x),y)dp_{\alpha\lambda_1+(1-\alpha)\lambda_2}(x,y) \geq \alpha \int \mathcal{L}(f_{\lambda_1}(x),y)dp_{\lambda_1}(x,y) + (1-\alpha)\int \mathcal{L}(f_{\lambda_2}(x),y)dp_{\lambda_2}(x,y)$. This proves concavity w.r.t the mixture weights $\lambda$, as desired.

$\square$

**Fact A.2.** *If $\mathcal{L}$ is $\ell_2^2$, then $\int \mathcal{L}(f_\lambda(x),y)p_\lambda(x,y)$ is concave in $\lambda$*

*Proof.* We assume $f_\lambda$ is the bayes optimal solution, hence we have the loss is the bayes error $\int_{\mathcal{X}} Var_{p_\lambda(y|x)}(y|x)p_\lambda(x)$. Note $p_\lambda(y|x) = \sum_{dp\in P} \lambda_p(x)p(y|x)$ where $\lambda_p(x) = \frac{\lambda_p p(x)}{\sum_{dp'\in P} \lambda_{p'} p'(x)} = \frac{\lambda_p p(x)}{p_\lambda(x)}$, and variance is concave w.r.t mixture weights. Thus we have

$$\int \mathcal{L}(f_\lambda(x),y)p_\lambda(x,y)$$

$$= \int_{\mathcal{X}} Var_{p_\lambda(y|x)}(y|x)p_\lambda(x)$$

$$\geq \int_{\mathcal{X}} \left( \sum_{dp\in P} \lambda_p(x) Var_{p(y|x)}(y|x) \right) p_\lambda(x)$$

$$= \int_{\mathcal{X}} \left( \sum_{dp\in P} \frac{\lambda_p p(x)}{p_\lambda(x)} Var_{p(y|x)}(y|x) \right) p_\lambda(x)$$

$$= \sum_{dp\in P} \lambda_p \int_{\mathcal{X}} Var_{p(y|x)}(y|x)p(x)$$

$$= \sum_{dp\in P} \lambda_p \int \mathcal{L}(f_p(x),y)p(x,y) \quad (3)$$

Note this inequality holds for an arbitrary set of distributions $P$. To now conclude concavity to the mixture weights, let $p_1 = p_{\lambda_1}, p_2 = p_{\lambda_2}$. Then by the above inequality we have $\int \mathcal{L}(f_{\alpha\lambda_1+(1-\alpha)\lambda_2}(x),y)dp_{\alpha\lambda_1+(1-\alpha)\lambda_2}(x,y) \geq \alpha \int \mathcal{L}(f_{\lambda_1}(x),y)dp_{\lambda_1}(x,y) + (1-\alpha)\int \mathcal{L}(f_{\lambda_2}(x),y)dp_{\lambda_2}(x,y)$. This proves concavity to mixture weights. $\square$

## B EMIXMAX GRADIENT DISCUSSION

**EMixMax is Unbiased:** We show that the expectation of $\nabla^{\text{EMixMax}}(\lambda)$ satisfies

$$\mathbb{E}_{D_p\sim dp\in P}\nabla^{\text{EMixMax}}(\lambda) = \sum_{dp\in P} \int \nabla_\lambda \lambda_p \mathcal{L}(f_\lambda(x),y)dp(x,y)$$

which is the gradient of the MixMax objective (Section 4.1).

**Fact B.1.** $\nabla^{\mathrm{EMixMax}}(\lambda)$ *is an unbiased gradient estimator for the MixMax objective (Equation 1), assuming* $|\nabla_\lambda \mathcal{L}(f_\lambda(x), y)|$ *is bounded and* $\mathcal{L}(f_\lambda(x), y)$ *is differentiable a.e (w.r.t* $\lambda$*).*

*Proof.* Note if $|\nabla_\lambda \mathcal{L}(f_\lambda(x), y)|$ is bounded, then by dominated convergence we have the derivative pulls into the integral for the MixMax objective (by the limit definition of derivatives, and the limit pulling in by dominated convergence)

$$\nabla_\lambda \sum_{dp \in P} \lambda_p \int \mathcal{L}(f_\lambda(x), y) dp(x, y) = \sum_{dp \in P} \int \nabla_\lambda \lambda_p \mathcal{L}(f_\lambda(x), y) dp(x, y)$$

Now note the expectation of EMixMax over sampling datasets of size $|D_p|$ i.i.d from $dp \in P$ is:

$$\mathbb{E}_{D_p \sim dp \in P} \sum_{dp \in P} \frac{1}{|D_p|} \sum_{(x,y) \in D_p} \nabla_\lambda \lambda_p \mathcal{L} \left( \sum_{dp \in P} \lambda_p \hat{f}_p(x), y \right)$$

$$= \sum_{dp \in P} \mathbb{E}_{D_p \sim dp \in P} \sum_{(x,y) \in D_p} \frac{1}{|D_p|} \nabla_\lambda \lambda_p \mathcal{L} \left( \sum_{dp \in P} \lambda_p \hat{f}_p(x), y \right)$$

$$= \sum_{dp \in P} \sum_{i=1}^{|D_p|} \frac{1}{|D_p|} \int \nabla_\lambda \lambda_p \mathcal{L} \left( \sum_{dp \in P} \lambda_p \hat{f}_p(x), y \right) dp(x, y)$$

$$= \sum_{dp \in P} \int \nabla_\lambda \lambda_p \mathcal{L} \left( \sum_{dp \in P} \lambda_p \hat{f}_p(x), y \right) dp(x, y)$$

Note by the previous equality this is the gradient of the MixMax objective (Equation 1).

$\square$

**Complexity of EMixMax** The EMixMax gradient computation in Algorithm 1 currently scales at $O(|P|(\sum_{dp} |D_p|))$. This can be prohibitive when the datasets are large. One can reduce the cost to $O(|P|^2)$ by doing mini-batch sampling at every step. In particular, for every optimization step and $p \in P$ one samples $B_p \sim D_p$ uniformly where $|B_p| = B$. One then replaces line 3 in Algorithm 1 with

$$l \leftarrow \sum_{dp \in P} \frac{\lambda_p}{B} \sum_{(x,y) \in B_p} \mathcal{L}(f_\lambda(x), y).$$

## C  EXPERIMENTAL SETUPS

### C.1  TOY EXPERIMENTS

For the binary classification task, We ran EMixMax with $D_i$ consisting of 10000 samples from each distribution, step size $\eta = 0.5$, and for 100 steps.

We ran EMixMax with $D_i$ again consisting of 10000 samples from each distribution, and $\eta = 0.0001$ for 100 steps.

### C.2  SEQUENCE MODELING

For the MixMax methods, letting $y$ denote a sequence, note the task was to model $p(y)$ and so had no covariates and hence had no covariate shift (as one can take $\mathcal{X}$ to be a singleton). Thus MixMax only needed functions for the token probabilities from each distribution, and we considered both the true probabilities (as the optimal baseline) and transformers trained on each Markov chain as the proxy optimal functions. The transformer used is GPTNeo Black et al. (2021) with 6 hidden states, 2 hidden

layers, 2 attention heads, 8 intermediate size, and with 12 max position embeddings. In both cases we used 200 samples per length to run EMixMax , keeping the other 600 to train the proxy model for $E^2$MixMax . The proxy model is trained for 20 epochs using AdamW with learning rates $0.01, 0.001$ and $0.0001$ (and otherwise default Pytorch hyperparameters).

We implemented DoReMi for finding mixture weights using the same transformer architecture as above, with the reference model being trained on a balanced dataset. The DoReMi reference and proxy models were again trained for 20 epochs using AdamW with learning rate $0.01, 0.001$ and $0.0001$ (and otherwise default Pytorch hyperparameters). The learning rate for the mixture weights was $0.1$. We also implemented group DRO with the same architecture, with the same model weights optimizer and mixture weights learning rate $0.1$. Furthermore, we tested varying minibatch sizes for the training of all models $(50, 100, 200)$, and number of steps used for group DRO $(150, 300, 600)$. In Figure 4 we reported the results for the hyperparameter settings with the lowest mean error for each method over 15 trials of randomly generating markov chains according to the varying magnitudes.

### C.3 TABULAR DATA

For running Gaussian kernel density estimation, we used the Scott method for bandwidth selection Scott (1979). For fitting the model on $E^2$MixMax with data reuse mixture weights, the reference models used for $E^2$MixMax with data reuse on the Income dataset, and the baseline of balancing the dataset, we implemented XGBoost with depths $6, 8, 10$, number of trees $100, 200, 300$, and learning rates $0.01, 0.1$ and reported the results for the hyperparameter setting with the best average accuracy over 5 trials with random train-test splits. For CelebA, the oracle accuracies and the reference models used in $E^2$MixMax with data reuse were always trained using depth 8, number of trees 200, and learning rate $0.1$; we found hyperparameter sweeping the models on individual groups had marginal impact, and anyways would only improve our results (the baseline of balancing always has a hyperparameter sweep). For training on the MixMax mixture or balanced mixture, we reweighed the loss for XGBoost using the sample weight parameter: for MixMax it is set to $\lambda_p/|D_p|$ for every datapoint in group $p$, and for balanced it is set to $1/|D_p|$. After normalization this gives the desired weighting of each group.

For the ensembled model results, we took the hyperparameters used for the retraining experiments and ran them for a separate 5 random train-test splits to evaluate the performance of ensembling the proxy model.

## D ADDITIONAL TABLES AND FIGURES

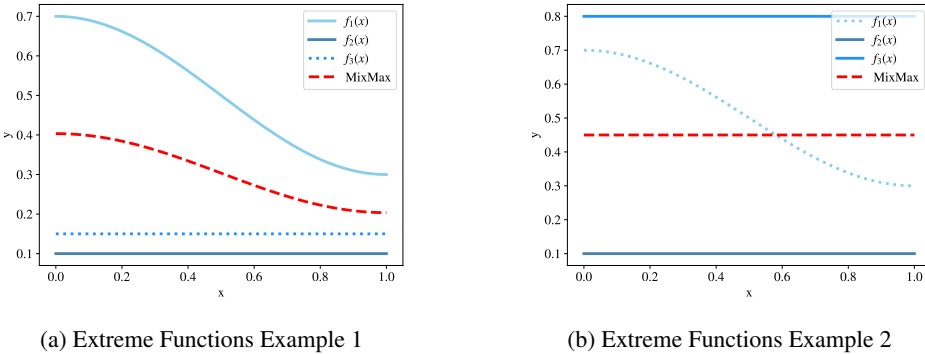

(a) Extreme Functions Example 1       (b) Extreme Functions Example 2

Figure 6: EMixMax for deterministic regression finds the average of the extreme functions. In general MixMax maximizes expected variance and here we plot the expected $y$ of the EMixMax mixture.

| Datasets | Features | Oracle (%) | Balanced (%) | E$^2$MixMax (%) |
|---|---|---|---|---|
| Inc-Race | All | 79.1($\pm$1.2) | 80.0($\pm$1.2) | 80.3($\pm$0.6) |
| | 2 | 71.2($\pm$0.1) | 70.7($\pm$0.2) | 70.6($\pm$0.3) |
| | 1 | 60.0($\pm$0.1) | 53.7($\pm$0.4) | 53.7($\pm$0.5) |
| Inc-Sex | All | 81.2($\pm$0.1) | 81.1($\pm$0.2) | 81.1($\pm$0.1) |
| | 2 | 72.8($\pm$0.1) | 71.4($\pm$0.2) | 71.6($\pm$0.1) |
| | 1 | 62.8($\pm$0.2) | 56.5($\pm$0.2) | **58.1($\pm$1.8)** |
| CelebA-Young | All | 77.0($\pm$0.1) | 75.6($\pm$0.2) | **76.0($\pm$1.0)** |
| | 10 | 68.3($\pm$0.3) | 62.5($\pm$0.3) | **64.8($\pm$1.6)** |
| | 5 | 63.4($\pm$0.3) | 53.8($\pm$0.2) | **57.0($\pm$0.6)** |
| CelebA-Pale Skin | All | 78.2($\pm$0.3) | 77.3($\pm$0.2) | 77.1($\pm$0.6) |
| | 10 | 67.6($\pm$0.1) | 66.8($\pm$0.2) | **67.1($\pm$0.3)** |
| | 5 | 62.4($\pm$0.2) | 59.5($\pm$0.3) | 59.4($\pm$0.3) |

Table 1: E$^2$MixMax with data reuse improved worst group accuracy over the baseline of balancing data for tabular dataset using XGBoost, in particular when more significant label shifts were present (see Figure 5b for a visualization). We present the average minimum accuracy over the groups from 5 trials with random training-test splits for E$^2$MixMax with data reuse, balancing by up-sampling, and the "oracle" accuracy where a model was trained for each group and is evaluated on the same group (as an example of optimal performance). Results are **bolded** if the better method's average performance was outside one standard deviation of the other method. We modified the datasets to study a variety of shifts by only including the first $N$ features.

| Datasets | Features | Oracle (%) | Retrained E$^2$MixMax (%) | Ensemble E$^2$MixMax (%) |
|---|---|---|---|---|
| Inc-Race | All | 79.1($\pm$1.2) | **80.3($\pm$0.6)** | 76.2($\pm$0.4) |
| | 2 | 71.2($\pm$0.1) | 70.6($\pm$0.3) | 70.4($\pm$0.3) |
| | 1 | 60.0($\pm$0.1) | **53.7($\pm$0.5)** | 53.3($\pm$0.3) |
| Inc-Sex | All | 81.2($\pm$0.1) | **81.1($\pm$0.1)** | 76.6($\pm$0.2) |
| | 2 | 72.8($\pm$0.1) | **71.6($\pm$0.1)** | 69.8($\pm$1.2) |
| | 1 | 62.8($\pm$0.2) | **58.1($\pm$1.8)** | 55.9($\pm$0.1) |
| CelebA-Young | All | 77.0($\pm$0.1) | 76.0($\pm$1.0) | 76.0($\pm$0.6) |
| | 10 | 68.3($\pm$0.3) | 64.8($\pm$1.6) | 62.6($\pm$2.7) |
| | 5 | 63.4($\pm$0.3) | 57.0($\pm$0.6) | 56.5($\pm$1.6) |
| CelebA-Pale Skin | All | 78.2($\pm$0.3) | 77.1($\pm$0.6) | 75.9($\pm$0.6) |
| | 10 | 67.6($\pm$0.1) | 67.1($\pm$0.3) | 67.2($\pm$0.2) |
| | 5 | 62.4($\pm$0.2) | 59.4($\pm$0.3) | 58.0($\pm$1.1) |

Table 2: Using E$^2$MixMax with data reuse mixture weights to train a new model performs better than ensembling the proxy models (following line 2 in Algorithm 1). We present the settings from Table 1, and similarly report the average minimum accuracy over the groups from 5 trials with random training-test splits for both methods of using MixMax mixture weights. Results are **bolded** if the better method's average performance was outside one standard deviation of the other method.

| Datasets | Features | Balanced | E$^2$MixMax |
|---|---|---|---|
| Inc-Race | All | 0.426($\pm$0.013) | 0.422($\pm$0.007) |
| | 2 | 0.557($\pm$0.001) | 0.559($\pm$0.003) |
| | 1 | 0.707($\pm$0.004) | **0.688($\pm$0.020)** |
| Inc-Sex | All | 0.403($\pm$0.004) | 0.405($\pm$0.005) |
| | 2 | 0.544($\pm$0.002) | **0.541($\pm$0.006)** |
| | 1 | 0.616($\pm$0.001) | **0.611($\pm$0.003)** |
| CelebA-Young | All | 0.495($\pm$0.002) | **0.488($\pm$0.015)** |
| | 10 | 0.644($\pm$0.002) | **0.614($\pm$0.019)** |
| | 5 | 0.706($\pm$0.002) | **0.670($\pm$0.007)** |
| CelebA-Pale Skin | All | 0.465($\pm$0.001) | 0.468($\pm$0.006) |
| | 10 | 0.594($\pm$0.001) | **0.588($\pm$0.005)** |
| | 5 | 0.647($\pm$0.001) | **0.641($\pm$0.005)** |

Table 3: E$^2$MixMax with data reuse improved worst group loss over the baseline of balancing data for tabular dataset using XGBoost. We present the average maximum loss over the groups from 5 trials with random training-test splits for E$^2$MixMax with data reuse and balancing by up-sampling. Results are **bolded** if the better method's average performance was outside one standard deviation of the other method. We modified the datasets to study a variety of shifts by only including the first $N$ features.

| Dataset | Features | Untuned E$^2$MixMax | Tuned E$^2$MixMax |
|---|---|---|---|
| Inc-Race | 1 | 53.7($\pm$0.5%) | **54.3($\pm$0.6%)** |
| Inc-Sex | 1 | 58.1($\pm$1.8%) | 58.4($\pm$1.8%) |
| CelebA-Young | 10 | 64.8%($\pm$1.6%) | 66.0%($\pm$2.1%) |
| | 5 | 57.0%($\pm$0.6%) | 57.0%($\pm$0.6%) |
| CelebA-Pale Skin | 5 | 59.4%($\pm$0.3%) | 59.4%($\pm$0.3%) |

Table 4: E$^2$MixMax with data reuse can sometimes be improved by doing a hyperparameter search over number of optimization steps $(20, 40, 60, 80)$. We present the settings from Table 1 where the gap between oracle accuracy and the balanced accuracy was more than $2\%$, and include the average minimum accuracy over groups over 5 trials of E$^2$MixMax with data reuse with further hyperparameter tuning. We **bolded** results where the tuned performance was outside one standard deviation of untuned.

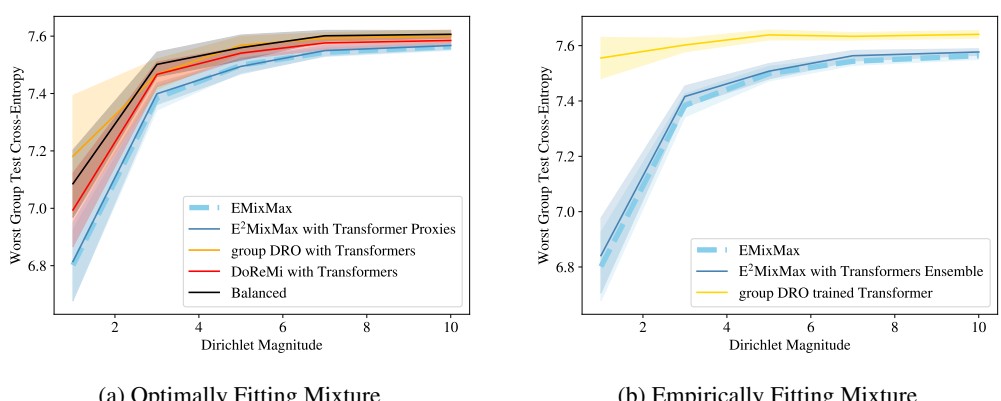

(a) Optimally Fitting Mixture

(b) Empirically Fitting Mixture

Figure 7: A reproduction of the experimental setup for Figure 7 but with fewer training samples (500 instead of 800). We still see that a model (empirically by ensembling transformers or optimally) fitted to E$^2$MixMax mixture weights performs better than DoReMi and group DRO mixture weights, and the model trained by group DRO. Furthermore, we still see that the improvement is even stronger when the distributions are less similar.

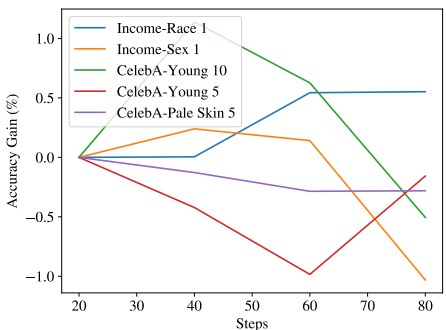

Figure 8: We see that $E^2$MixMax with data reuse can benefit from early stopping for several datasets, as seen by the peaks in performance, however these trends are statistically weak given the standard deviations reported in Table 4. Here we plot the worst group accuracy gain over running $E^2$MixMax with data reuse for 20 steps when running $E^2$MixMax with data reuse for $20, 40, 60, 80$ steps. Further hyperparameter details are described in Appendix C. This is done for the datasets observed to have big label shifts, described in Table 1

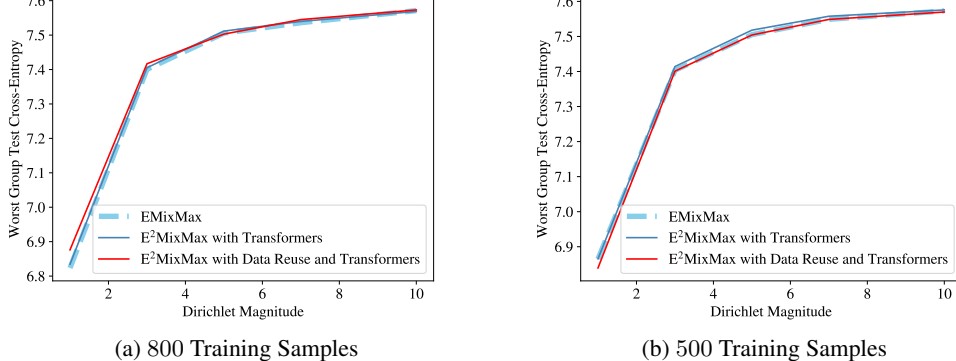

(a) 800 Training Samples

(b) 500 Training Samples

Figure 9: $E^2$MixMax with data reuse is not much worse than $E^2$MixMax , and can be slightly better when one has fewer training samples. Here we consider the experimental setup from Figure 4a for fitting the various MixMax mixture weights optimally, and present results for when the same 800 or 500 training points are used to fit the proxy models and run EMixMax giving the data reuse version (instead of the $600 - 200$ and $300 - 200$ split for proxy training and MixMax).

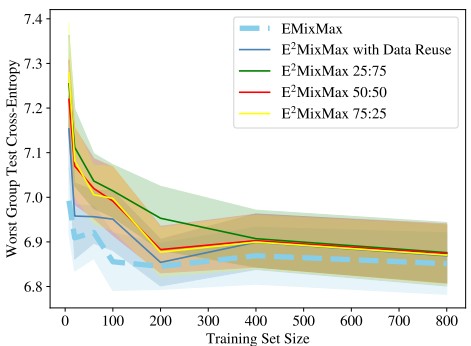

Figure 10: Various empirical MixMax approaches perform comparably for worst group cross-entropy, with E²MixMax with data reuse performing closest to EMixMax (with ground truths) in the small training set regime. Here we plot the mean and $95\%$ confidence interval (over $45$ trials of sampling new sets of Markov-chains from the symmetric Dirichlet distribution with magnitude 1) of the worst group cross-entropy of the Bayes optimal function defined by the methods's mixture weights, changing the number of training samples per length.

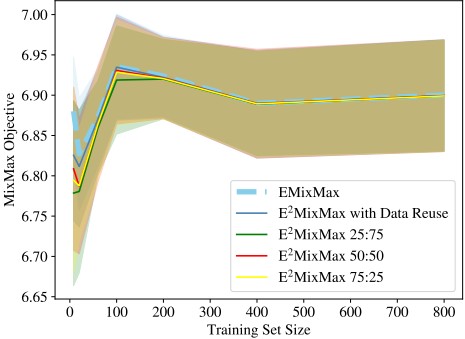

Figure 11: Various empirical MixMax approaches perform comparably in maximizing the MixMax objective, with E²MixMax with data reuse performing closest to EMixMax (which uses ground truths) in the small training set regime. Here we plot the mean and $95\%$ confidence interval (over $45$ trials of sampling new sets of Markov-chains from the symmetric Dirichlet distribution with magnitude 1) of the MixMax objective value for the methods's mixture weights, changing the number of training samples per length.

