# OpenReview forum: "MixMax: Distributional Robustness in Function Space via Optimal Data Mixtures"
_ICLR.cc/2025/Conference — ICLR 2025 Poster_

### Official Review · Reviewer_MsQo · 2024-10-27

**Soundness:** 3
**Presentation:** 3
**Contribution:** 2
**Rating:** 5
**Confidence:** 4

**Summary:**

The paper proposes a reparameterization technique for the group DRO problem to address the non-convexity of the parameters, under the most popular machine learning backgrounds. This reparameterization from parameter space to bounded function space admits a generalized minimax theorem. Moreover, the paper proves that group DRO can be decomposed into a convex optimization problem followed by a risk minimization problem for the frequently-used cross-entropy and mean squared error. Comprehensive experimental results align closely with the theoretical guarantee.

**Strengths:**

1. The idea of reparameterization seems novel in the context of group DRO.
2. The experiments are comprehensive in terms of the choice of datasets and the diverse ways of presentation.
3. The paper is written clearly, especially for the theoretical results.

**Weaknesses:**

There are multiple disadvantages, mainly regarding the methodology itself, together with its relevance to the existing literature on (empirical) group DRO.

1. The main contribution of this paper (Theorem 3.1) seems a direct generalization of Sion's minimax theorem by transforming the topology in the function space.
2. The key discovery is that group DRO can be solved by fitting a specific mixture distribution. However, this seems like a straightforward idea in the optimization community (Nemirovski et al., 2009, Section 3.2) and recent works on group DRO (Zhang et al., 2023; Yu et al., 2024) also use this property to reformulate the optimization problem. I understand that the second step in this paper, i.e., transforming the minimax object into a concave optimization problem of the distributional weights, is different from the ways in the papers I cited, but the first step (supported by the main theorem), has already been exploited in a similar way
3. The reparameterization enables line 2 of Algorithm 1, which allows the possibility of iterative training and the whole MixMax schema. However, I am a little skeptical about parameterizing $f_\lambda(\cdot)$ in this form. It seems that MixMax exerts a constraint on the function class in the group DRO objective, which is not satisfactory. What happens if $f_\lambda(\cdot)$ can not be expressed via the closed form shown in the paper?
4. As an optimization paper in ICLR, the authors should consider adding complexity analysis of the proposed algorithm, either the computation complexity of Empirical MixMax, or the sample complexity of Population MixMax. From my understanding, Algorithm 1 needs to compute the empirical risk among each data group per iteration. This seems too expensive for modern machine learning or large-scale optimization tasks. Although the experiments verify the effectiveness of MixMax over other group DRO approaches, theoretical justifications still need to be presented in an optimization paper.
5. The original form of group DRO involves risk function in the form of expectation, which is intractable in reality. That's why stochastic approximation algorithms prevail. However, as it requires computing the risk function for each group in Algorithm 1, I was wondering if MixMax can be directly applied to group DRO (at population level). If not, then the main contribution of this paper should be the reparameterization of empirical group DRO, rather than group DRO.
6. The baselines used in the experiments could be improved. The algorithm proposed by Sagawa et al. (2020) suffers from a suboptimal complexity, which may cause unfair comparisons in Figure 4. The authors could try adopting the near-optimal algorithms of Zhang et al. (2023, Algorithm 1 and 2), whose complexity is $|P|$ times lower than that of Sagawa et al. (2020).
7. Both the algorithm and the experiments target empirical group DRO problems. However, only group DRO algorithms are compared. It would be more persuasive to compare with empirical group DRO algorithms (Yu et al., 2024) since the latter could use the problem structure to further improve the complexity.

[1] Nemirovski, A., Juditsky, A., Lan, G., and Shapiro, A. Robust stochastic approximation approach to stochastic programming. SIAM Journal on Optimization, 2009.

[2] Zhang, L., Zhao, P., Yang, T., and Zhou, Z.-H. Stochastic approximation approaches to group distributionally robust optimization. In NeurIPS, 2023.

[3] Yu, D., Cai, Y., Jiang, W., and Zhang, L. Efficient Algorithms for Empirical Group Distributionally Robust Optimization and Beyond. In ICML, 2024.

[4] Sagawa, S., Koh, P. W., Hashimoto, T. B., and Liang, P. Distributionally robust neural networks for group shifts: On the importance of regularization for worst-case generalization. In ICLR, 2020.

**Questions:**

I would appreciate it if the authors were willing to answer some of the questions I raised in the weaknesses above.

---

> ### Author Response · Authors · 2024-11-18
> **Response to Reviewer MsQo**
>
> We thank the reviewer for their time and feedback! We discuss the questions raised by the reviewer below, and have incorporated a lot of their feedback, strengthening our paper. In particular, we believe DoReMi implements a variant of Zhang et al’s algorithm. We now state this in the paper. Because we already compare MixMax to DoReMi, we believe this addresses the question of comparisons to near-optimal convex optimization algorithms, but we are happy to discuss this further with the reviewer.
>
> We have also added detail on the computational complexity of EMixMax. Furthermore, in the discussion below we clarify the differences with past parametric convex-concave optimization methods and our setting, and now cite the mentioned papers. We hope the reviewer considers raising their score given these changes.
>
> > The main contribution of this paper (Theorem 3.1) seems a direct generalization of Sion's minimax theorem by transforming the topology in the function space.
>
> To clarify how we proved Theorem 3.1, note Sion’s minimax theorem requires a topology that is compact over the space we are minimizing. In the case of optimization over $\mathbb{R}^n$, this is satisfied by restricting the set to be bounded and closed. However, for $L^{\infty}$, a bounded and closed subset is not compact in the respective $L^{\infty}$ norm; in fact, the subset would not be compact in any $L^p$ norm (as mentioned in the paper). Our first contribution in the paper is making the connection that the weak*-topology over bounded functions is compact (by Banach-Alaoglu’s theorem), and furthermore, the group DRO objective is continuous in the weak*-topology (shown by the method of test functions and $L^p$ space duality). Hence, Sion’s minimax theorem can apply over the set of bounded functions by the existence of a suitably topology.
>
> To the best of our knowledge, no prior minimax theorem over bounded functions existed. Furthermore, our paper provides several other contributions to the theory and practice of group DRO over bounded functions which builds on this first result of saddle points existing (see discussion to questions below).
>
> > The key discovery is that group DRO can be solved by fitting a specific mixture distribution. However, this seems like a straightforward idea in the optimization community (Nemirovski et al., 2009, Section 3.2) and recent works on group DRO (Zhang et al., 2023; Yu et al., 2024) also use this property to reformulate the optimization problem. I understand that the second step in this paper, i.e., transforming the minimax object into a concave optimization problem of the distributional weights, is different from the ways in the papers I cited, but the first step (supported by the main theorem), has already been exploited in a similar way
>
> Thank you for the references, we have added them to the related work section. To clarify the differences in setup, all of the above papers consider convex-concave optimization over a subset of $\mathbb{R}^n$ where saddle-points were already known to exist, and devise algorithms to jointly converge to the saddle point (in both the minimization parameter and maximization parameter). The key difference is our optimization is over bounded functions, where apriori saddle-points were not known to exist. Our first step was to prove saddle-points exist for our problem.
>
> To briefly clarify the “second step” described by the reviewer, another difference between our setup and the cited papers is that even if a saddle-point exists for our problem, we cannot readily evaluate the function space gradients of our objective (e.g., the literature on gradient boosting to approximate these gradients). Hence, we can not directly apply the gradient based methodology provided in previous papers to obtain our saddle-points. We bypass this by showing that for two common loss functions, the mixture for the saddle-point satisfies a concave maximization over a finite-dimensional simplex. We elaborate on how we show this in response to the next question.

---

> > ### Author Response · Authors · 2024-11-18
> > **Response to Reviewer MsQo Continued**
> >
> > > The reparameterization enables line 2 of Algorithm 1, which allows the possibility of iterative training and the whole MixMax schema. However, I am a little skeptical about parameterizing fλ(⋅) in this form. It seems that MixMax exerts a constraint on the function class in the group DRO objective, which is not satisfactory. What happens if fλ(⋅)can not be expressed via the closed form shown in the paper?
> >
> > For the losses studied in this paper, cross-entropy and $\ell_2^2$, the minimizer over all bounded functions of the mixture distribution always follows the parameterized form in line 2 of Algorithm 1 (see Section 4.1). To elaborate, the minimizer for a distribution over all bounded functions is the Bayes optimal function for that distribution. For cross-entropy, the Bayes optimal function is the conditional $p(y|x)$ for that distribution. By definition of conditional distributions and mixture distributions, the Bayes optimal function of a mixture distribution follows the parametric formula in line 2 of Algorithm 1. Similarly, for $\ell_2^2$, the Bayes optimal function is $E_{p_{\lambda}(y|x)} y$, and once again by the definition of expectation and conditional distributions, we have the Bayes optimal function of a mixture follows line 2 of Algorithm 1.
> >
> > > As an optimization paper in ICLR, the authors should consider adding complexity analysis of the proposed algorithm, either the computation complexity of Empirical MixMax, or the sample complexity of Population MixMax. From my understanding, Algorithm 1 needs to compute the empirical risk among each data group per iteration. This seems too expensive for modern machine learning or large-scale optimization tasks. Although the experiments verify the effectiveness of MixMax over other group DRO approaches, theoretical justifications still need to be presented in an optimization paper.
> >
> > Thanks for pointing out the cost of computing the gradients in Algorithm 1. Briefly, the current complexity is $O(|P|\sum_{dp \in P} |D_p|)$. When the datasets are large this is dominated by the size of the datasets. One can reduce this to $O(|P|^2)$ by sampling a fixed size mini-batch from every group to compute the gradient. We now include a short note on this in the main body together with discussion and an example of a stochastic gradient estimator in Appendix B.
> >
> > Thanks again for pointing this out, we think this has really strengthened the paper!
> >
> > > The original form of group DRO involves risk function in the form of expectation, which is intractable in reality. That's why stochastic approximation algorithms prevail. However, as it requires computing the risk function for each group in Algorithm 1, I was wondering if MixMax can be directly applied to group DRO (at population level). If not, then the main contribution of this paper should be the reparameterization of empirical group DRO, rather than group DRO.
> >
> > To clarify, our theory for MixMax is for population group DRO (we did not restrict the integral to be a finite sum), and our experiments evaluate test group DRO performance. The confusion may be coming from Section 4 and the experiments, where we optimize an empirical approximation of the population MixMax objective (a monte-carlo estimate of the expectation), but note we always estimate the solution’s population group DRO performance in the experiments (by using a held-out test set). Our experimental results show EMixMax is effective at improving test group DRO performance.

---

> > > ### Author Response · Authors · 2024-11-18
> > > **Response to Reviewer MsQo Continued**
> > >
> > > > The baselines used in the experiments could be improved. The algorithm proposed by Sagawa et al. (2020) suffers from a suboptimal complexity, which may cause unfair comparisons in Figure 4. The authors could try adopting the near-optimal algorithms of Zhang et al. (2023, Algorithm 1 and 2), whose complexity is |P| times lower than that of Sagawa et al. (2020).
> > >
> > > Thanks for the additional reference, we have added it to the paper. Firstly, we believe DoReMi in fact implements Algorithm 1 in Zhang et al with additional hyperparameters. DoReMi samples a mini-batch from each group at every step similar to Zhang et al (though, in our setting, with a very small probability $\leq 3 (⅔)^{\text{batch size}} \sim 10^{-26}$, DoReMi will sample no data points from some group) and then construct unbiased estimates of the gradient for each group and weigh them similar to Zhang et al. From our reading, the major difference is that DoReMi hyper-parameter tunes additional regularizing hyper parameters for non-convex optimization with transformers. We now state the following at the beginning of Section 6.2:
> > >
> > > *“DoReMi (Xie et al., 2024) which follows the near sample optimal algorithm of Zhang et al. (2024)
> > > for convex group DRO”*
> > >
> > > Secondly, on the potential issue of needing more samples for some methods in the synthetic non-convex parametric experiments (Section 6.2), in Figure 7 in the Appendix we found the results had already converged in samples (results were the same for fewer samples).
> > >
> > > Finally, our main real-world application was to non-parametric models where the mentioned algorithms do not apply (for lack of parameters and gradients). In that setting the only known baseline was balancing the dataset.
> > >
> > > > Both the algorithm and the experiments target empirical group DRO problems. However, only group DRO algorithms are compared. It would be more persuasive to compare with empirical group DRO algorithms (Yu et al., 2024) since the latter could use the problem structure to further improve the complexity.
> > >
> > > Thanks for the additional reference, we have added it to the paper. On targeting empirical group DRO problems, our theory for MixMax is in fact for population group DRO, and our experiments evaluate test group DRO performance (which approximates the population group DRO performance): see discussion to previous questions. On baselines for population group DRO, as mentioned earlier, we believe we had in fact implemented a variant of the near sample optimal population group DRO algorithm of Zhang et al. for our synthetic parametric experiments. Finally, our main (real-world) experiments were for non-parametric learning, where parametric methods (such as those in the suggested papers) do not apply.

---

> > > > ### Comment · Reviewer_MsQo · 2024-11-19
> > > > **Official Comment by Reviewer MsQo**
> > > >
> > > > Thank you for your detailed and thoughtful response. I appreciate the effort the authors put into addressing my concerns. Upon evaluating the revised manuscript and reflecting on the feedback from other reviewers, I am confident that the authors have made meaningful improvements. As a result, I have decided to adjust my score upward to acknowledge these positive changes.

---

### Official Review · Reviewer_DjaM · 2024-11-02

**Soundness:** 2
**Presentation:** 2
**Contribution:** 2
**Rating:** 6
**Confidence:** 2

**Summary:**

This paper aims to provide a solution to the problem of group distributionally robust optimization (DRO). It considers the class of all bounded functions to address this issue. The group DRO problem is a specific case of the broader DRO problem, where the set of distributions is finite and consists of predefined distributions.

The paper first demonstrates that group DRO over the space of bounded functions satisfies a minimax theorem. Additionally, it claims that when the loss function is either cross-entropy or mean squared error, the problem can be transformed into a convex optimization problem, which subsequently leads to a classical risk minimization problem.

**Strengths:**

The issue is both intriguing and significant in the literature.

**Weaknesses:**

The theoretical proofs are quite brief and unclear. It would be beneficial to discuss and clarify the proofs step by step.

**Questions:**

1- I believe the VC dimension of the class of bounded functions is unbounded. Could this present a challenge for solving real-world problems?

2- It is claimed that in Equation 1 the objective is concave with respect to $\lambda$. However, the convexity of $f_\lambda$ with respect to $\lambda$ is not clear. Although it is true that, if the class is rich enough, it can approximate the error of $p_\lambda(y|x)$, it is not exactly equal to this function.

3-To prove that the function $f_\lambda$ is concave with respect to $\lambda$, it must be shown that $f_{\alpha \lambda_1 + (1 - \alpha) \lambda_2} \geq \alpha f_{\lambda_1} + (1 - \alpha) f_{\lambda_2}$. However, in Equation 3, this is not demonstrated. Instead, concavity with respect to $p$ is proven in that equation.

---

> ### Author Response · Authors · 2024-11-18
> **Response to Reviewer DjaM**
>
> We thank the reviewer for their time and feedback! We discuss the weaknesses and questions raised by the reviewer below, and believe we have addressed the reviewer’s primary concern regarding the proof of concavity by adding a final step to the proof which was implicit before. We ask the reviewer to consider raising their score.
>
> > The theoretical proofs are quite brief and unclear. It would be beneficial to discuss and clarify the proofs step by step.
>
> We clarify the proofs of concavity in our response to the questions below, and have added more detail to those proofs in the paper.
>
> > I believe the VC dimension of the class of bounded functions is unbounded. Could this present a challenge for solving real-world problems?
>
> Yes, in theory this is true, by using a classical analysis the VC dimension of bounded functions would show up as an error in our ability to learn the $\hat{f_p}$. However, our experiments suggest that MixMax is an improvement on baselines in real-world settings when we learn over highly expressive model classes (e.g., XGBoost).
>
> > It is claimed that in Equation 1 the objective is concave with respect to λ. However, the convexity of $f_λ$ with respect to $\lambda$ is not clear. Although it is true that, if the class is rich enough, it can approximate the error of pλ(y|x), it is not exactly equal to this function.
>
> To clarify, we showed the MixMax objective is concave in $\lambda$: see our response to the next question for an elaboration on the proof. This is distinct from the Bayes optimal function $f_{\lambda}$ being concave or convex in $\lambda$.
>
> > To prove that the function $f_{\lambda}$ is concave with respect to $\lambda$, it must be shown that $f_{αλ_1+(1−α)λ_2}≥ αf_{\lambda_1}+(1−α)f_{\lambda_2}$. However, in Equation 3, this is not demonstrated. Instead, concavity with respect to p is proven in that equation.
>
> Our proof of concavity to the distribution $dp$ implies the concavity condition you stated for the MixMax objective, but this was not explicit in the writing. We have now added this step to the proofs in the appendix.
>
> Briefly, we had shown $MixMax(\sum \lambda_i p_{i}) \geq \sum \lambda_i MixMax(p_i)$. Defining $p_1 = p_{\lambda_1}$ and $p_2 = p_{\lambda_2}$, this implies $MixMax(p_{\alpha \lambda_1 + (1-\alpha)\lambda_2}) = MixMax(\alpha p_{\lambda_1} + (1-\alpha)p_{\lambda_2}) \geq \alpha MixMax(p_{\lambda_1}) + (1-\alpha)MixMax(p_{\lambda_2})$. Hence the Mixmax objective is concave in the mixture weights.

---

> ### Comment · Reviewer_DjaM · 2024-11-21
> **Response to Authors**
>
> Thank you for your response.
>
> Dear authors, I appreciate your answer and your clarification regarding the concavity of the objective function when $f_\lambda$ is equal to the Bayes optimal solution.
>
> I would like to clarify that my comment about the proofs being short and unclear is not limited to the proof of concavity that you mentioned. For a paper that primarily contributes theoretical insights, I find your mathematical proofs to be brief and compressed, lacking sufficient detail (for example, in the proof of Theorem 3.1).
>
> Additionally, your explanation regarding the VC dimension of the class of bounded functions did not convince me. If the VC dimension is unbounded, it suggests that the model may not generalize well; specifically, the empirical minimax may not converge around its expected value uniformly. The reason you may be obtaining good experimental results could be because you have parameterized the function class in a way that makes it more restricted, which in turn reduces the VC dimension. However, this implies that your class may not be general enough to match the performance of the Bayes optimal classifier, as you assumed in your proofs. Consequently, as I mentioned in my second question, this undermines your proof of the concavity of the objective function. While you have demonstrated that the objective function is concave when $f_\lambda$ equals the Bayes optimal solution, it must also be shown that your new restricted class can achieve this solution.

---

> > ### Author Response · Authors · 2024-11-22
> > **Response to Reviewer**
> >
> > We thank the reviewers for the clarifications of their concerns! We’ve made further edits to the paper draft following the discussion below, and look forward to the reviewer’s response.
> >
> > > I would like to clarify that my comment about the proofs being short and unclear is not limited to the proof of concavity that you mentioned. For a paper that primarily contributes theoretical insights, I find your mathematical proofs to be brief and compressed, lacking sufficient detail (for example, in the proof of Theorem 3.1).
> >
> > Sorry for missing that the comment on clarity was also regarding Theorem 3.1. Could the reviewer identify the specific parts of the proof of Theorem 3.1 that they think are unclear or vague? We will try our best to update the manuscript to clarify these points in the proof.
> >
> > > Additionally, your explanation regarding the VC dimension of the class of bounded functions did not convince me. If the VC dimension is unbounded, it suggests that the model may not generalize well; specifically, the empirical minimax may not converge around its expected value uniformly. The reason you may be obtaining good experimental results could be because you have parameterized the function class in a way that makes it more restricted, which in turn reduces the VC dimension. However, this implies that your class may not be general enough to match the performance of the Bayes optimal classifier, as you assumed in your proofs. Consequently, as I mentioned in my second question, this undermines your proof of the concavity of the objective function. While you have demonstrated that the objective function is concave when ( f_\lambda ) equals the Bayes optimal solution, it must also be shown that your new restricted class can achieve this solution.
> >
> > Sorry for misunderstanding the question earlier. We thank the reviewer for bringing up some important points which have helped us improve the clarity of the draft. We list the changes we have made to the manuscript below:
> >
> >
> > 1) We clarify that the empirical MixMax methods we propose are stochastic gradient methods (for the MixMax objective), not an empirical risk minimization. Hopefully this clarifies why we did not prove the concavity of the empirical MixMax objective that we had presented in the first draft. We have rewritten Section 4.2 accordingly, and we think it is much stronger and clearer.
> >
> > 2) In particular we also provide a proof in Appendix B that EMixMax is an unbiased gradient estimator.
> >
> > 3) We now also emphasize the limitations of our practical gradient estimator (E$^2$MixMax) in Section 4.2. In particular, that we do not prove what its bias is (or how errors in the estimation of the components $\hat{f}_p$ propagate to $\lambda$), but rather rely on empirical findings. Specifically, our empirical findings are that it performs on par with unbiased gradient estimators (Figure 4), and improves the only known group DRO baseline for non-parametric learning (Section 6.3).
> >
> > 4) Finally, we emphasize in the introduction (lines 70-72) that our theory assumes idealistic access assumptions to optimize the MixMax objective, and that we only experimentally justify the use of our more practical E$^2$MixMax gradient estimator.
> >
> >
> >
> > There are still some parts of your comment we are not sure we fully understand, which we list below:
> >
> > 1) We do not understand how the component functions dictate the concavity of MixMax; it is an objective only defined by the distributions. We interpreted this comment as confusion regarding our gradient estimators for MixMax, but please correct us if we are wrong.
> >
> >
> > 2) We do not entirely follow the comment on VC dimension. We interpreted this as trying to use the VC dimension to understand the error in the E$^2$MixMax gradient estimator (i.e., understanding $\hat{f}_p$). Please let us know if we are wrong.

---

> > > ### Comment · Reviewer_DjaM · 2024-11-23
> > > **Response to Authors**
> > >
> > > Concerning my concerns about the clarity of proofs, the proof for Theorem 3.1 spans approximately half a page. The initial three paragraphs are largely descriptive rather than mathematical.
> > >
> > > Regarding my concerns about VC dimension,  I believe you understand my points based on your comments. It seems that your theory applies only in an asymptotic scenario. When handling samples, you adjust your function class to a specific set of classifiers, which is not as comprehensive as the entire class of all bounded functions. As a result, your theory does not hold for this modified class. Therefore, I feel there is an inconsistency between your theory and the experimental results.
> > >
> > > I must admit that I am not entirely satisfied with the paper. However, based on your explanations, I believe it leans more toward acceptance than rejection, although it requires some general modifications to clarify the proofs. Therefore, I am raising my score accordingly.

---

> > > > ### Author Response · Authors · 2024-11-24
> > > > **Response to Reviewer**
> > > >
> > > > We really appreciate the time and feedback the reviewer has given for our paper. As suggested, we have expanded our proof of Theorem 3.1. We describe the changes in more detail below.
> > > >
> > > > > Concerning my concerns about the clarity of proofs, the proof for Theorem 3.1 spans approximately half a page. The initial three paragraphs are largely descriptive rather than mathematical.
> > > >
> > > > Upon reviewing our proof we agree several parts described a series of calculations without providing those calculations. We have now added those calculations which were:
> > > >
> > > > 1) Calculations showing $\sup_{dp \in P} = \sup_{dp \in Conv(P)}$ for our objective
> > > > 2) Calculations involved with the use of the test function $g$ to conclude pointwise convergence
> > > > 3) Calculations involved with deducing continuity from weak* convergence (via pointwise convergence)
> > > >
> > > > We believe this has improved the clarity of our proof, and thank the reviewer for the comment on avoiding descriptive arguments.
> > > >
> > > > > Regarding my concerns about VC dimension, I believe you understand my points based on your comments. It seems that your theory applies only in an asymptotic scenario. When handling samples, you adjust your function class to a specific set of classifiers, which is not as comprehensive as the entire class of all bounded functions. As a result, your theory does not hold for this modified class. Therefore, I feel there is an inconsistency between your theory and the experimental results.
> > > >
> > > >
> > > > We agree with the reviewer on our theory no longer holding for our experiments using E$^2$MixMax. We believe one might be able to prove error bounds between E$^2$MixMax and EMixMax (the unbiased estimator for MixMax), as that is consistent with our experimental findings (Figure 4). Currently we do not know how to make that argument, and hope future work is able to tackle this problem or possibly provide even better gradient estimators than E$^2$MixMax.
> > > >
> > > > We again thank the reviewer for all their feedback, our paper has truly become a lot stronger because of it.

---

### Official Review · Reviewer_8SGa · 2024-11-04

**Soundness:** 4
**Presentation:** 3
**Contribution:** 3
**Rating:** 8
**Confidence:** 4

**Summary:**

The paper formalizes the DRO problem in the case of bounded functions and shows that a solution is an optimal data mixture. In case of cross-entropy and L2 losses the problem can be further simplified to a single maximization of a concave function. An empirical version of the problem is further presented even in the case where the Bayes optimal functions are unknown. Several experiments on both synthetic and real data demonstrate that the empirical version of the MixMax improves over previous group DRO baselines in parametric cases as well as over non parametric data balancing algorithms.

**Strengths:**

- The major theoretical result that under certain conditions the DRO problem is equivalent to a single maximization over a concave objective function and that a solution can be achieved by fitting an optimal data mixture.

- The authors also provide a practical algorithm that computed the empirical MixMax solution.

**Weaknesses:**

- $Emperical^2$ $MixMax$ relies on an accurate estimation of $\hat{f}_p(x)$, which in practice requires to optimally fit a model on every distribution, including performing a HP search by cross validation. This might be computationally prohibitive in some practical cases. Could you discuss computational trade-offs or potential approximations that could be used in resource-constrained settings? How those approximations would impact the  $Emperical^2$ $MixMax$ estimate?

- In some of the experiments where a Mixmax method was compared to an alternative method, the MixMax requires more compute which makes the comparison less fair. (e.g. experiment 6,3). It would be helpful to have experiments that control for computational budget across methods.

**Questions:**

- For the experiment described in 6.3, which strategy was used to upsample the minority class ? Was the training loss reweighed in order to de-bias the loss estimation ?

- Two different ways of using the MixMax weights were described, either retraining a model on the weighted dataset or reweighing models at inference time. Do you have a sense of how those 2 approaches compare in practice under different settings ?

---

> ### Author Response · Authors · 2024-11-18
> **Response to Reviewer 8SGa**
>
> We thank the reviewer for their time and interesting questions! We discuss the weaknesses and questions raised by the reviewer below. Your comments have helped us improve the paper. In particular, we have now added experiments directly comparing the two ways of using MixMax weights.
>
> > Emperical2MixMax relies on an accurate estimation of $\hat{f_p}(x)$, which in practice requires to optimally fit a model on every distribution, including performing a HP search by cross validation. This might be computationally prohibitive in some practical cases. Could you discuss computational trade-offs or potential approximations that could be used in resource-constrained settings? How those approximations would impact the Emperical2 MixMax estimate?
>
> We agree that understanding how different approximations of $f_p(x)$ impact mixture finding is interesting and important to study. Towards this, in figure 10 (in the Appendix) we studied how obtaining the approximations $\hat{f_p}(x)$ by training with fewer samples (and hence with less compute) affected the final mixture performance. We found performance converged quickly with samples: training with ¼ of the dataset gave mixtures comparable to using the whole dataset. We hope future work characterizes more broadly under what approximation errors to $f_p$ we can recover close to optimal mixture weights. For example, what happens if we use smaller models instead of less samples to save compute?
>
> > In some of the experiments where a Mixmax method was compared to an alternative method, the MixMax requires more compute which makes the comparison less fair. (e.g. experiment 6,3). It would be helpful to have experiments that control for computational budget across methods
>
> This is a great point. The only baseline applicable in Section 6.3 (non-parametric models) is data balancing. It is not fair to compare compute in this setting, as data balancing uses 0 compute to return its mixture weight, but it can be arbitrarily suboptimal.
>
> For the parametric problems in Section 6.2, there are algorithmic baselines available and MixMax uses comparable compute to them. For example, both DoReMi and Group DRO require training one model on all the data, whereas we require training three models on one-third of the data each: up to hyperparameter tuning the number of training steps (e.g., for early-stopping), this is the same cost.
>
> > For the experiment described in 6.3, which strategy was used to upsample the minority class ? Was the training loss reweighed in order to de-bias the loss estimation?
>
> Thanks for raising this question, this is in fact a typo. We reweighed the training loss for XGBoost according to the MixMax weights or balanced weights. This is now described in more detail in Appendix C.3.
>
> > Two different ways of using the MixMax weights were described, either retraining a model on the weighted dataset or reweighing models at inference time. Do you have a sense of how those 2 approaches compare in practice under different settings ?
>
> This is an interesting question! We have added results comparing the two ways of using the MixMax weights for the tabular experiments, shown in Table 2 in the Appendix. We observed training a new model performs better or as good as ensembling the proxies across all the tabular experiments. Retraining comes with additional training cost, while ensembling comes with more inference cost, so we leave it to practitioners to decide if the improved performance offsets the varying costs.

---

> ### Author Response · Authors · 2024-11-23
> **Looking Forward to Further Discussion**
>
> We look forward to your thoughts on our response. Let us know if there is anything more we can do to address your comments!
>
> Thanks again for you time,
>
> Authors

---

### Official Review · Reviewer_13Yx · 2024-11-04

**Soundness:** 3
**Presentation:** 4
**Contribution:** 3
**Rating:** 8
**Confidence:** 2

**Summary:**

This paper provides a reparameterization of the group DRO objective from the parameter space to function space under the assumption that the hypothesis class for the predictor contains the set of the bayes optimal classifiers. Under this reparameterization, they introduce an algorithm to optimize this objective. Empirical results show the proposed algorithm outperforms groupDRO.

**Strengths:**

1)The problem is important and relevant to OOD generalization.

2)The proposed method is novel in its approach and theoretically justified.

3)Experimental results show that the proposed method outperforms groupDRO.

4)The paper is well written.

**Weaknesses:**

See questions.

**Questions:**

1)Under the assumption that the bayes optimal predictor is a subset of the set of hypothesis, wouldn't the same f minimize the objective for any dp in the set P? If this is the case, this weakens the theoretical contribution.

---

> ### Author Response · Authors · 2024-11-18
> **Response to Reviewer 13Yx**
>
> We thank the reviewer for their time and feedback! Below we answered the question raised by the reviewer. If there are no additional concerns we ask the reviewer to consider raising their score.
>
> > Under the assumption that the bayes optimal predictor is a subset of the set of hypothesis, wouldn't the same f minimize the objective for any dp in the set P? If this is the case, this weakens the theoretical contribution.
>
> The Bayes optimal function is not the same for every distribution, hence a single f won’t minimize every distribution. For a specific example, consider binary classification over a single input.  Take $P = \{p_1,p_2\}$ where $p_1(y=1) = 0.9$, $p_2(y=1) = 0.2$. Then the Bayes optimal function are $f_1(1) = 0.9$ and $f_2(1) = 0.2$ and MixMax would give $f_{\lambda}(1) = 0.5$ with optimal MixMax weights being $\lambda = [3/7, 4/7]$. Notice $f_{\lambda}$ is not optimal for $p_1$ or $p_2$.

---

> ### Author Response · Authors · 2024-11-23
> **Looking Forward to Further Discussion**
>
> We look forward to your thoughts on our response. Let us know if there is anything more we can do to address your comments!
>
>
>
> Thanks again for you time,
>
> Authors

---

> > ### Comment · Reviewer_13Yx · 2024-12-03
> >
> > Thank you for your response, I think this paper makes a good contribution and have raised my score to reflect so.

---

### Author Response · Authors · 2024-11-18
**Summary of Reviews and Responses**

We thank the reviewers for their time! The reviewers’ feedback has helped us strengthen the paper. We have made changes to the manuscript, highlighted in blue for ease of reading.

We were glad to see that collectively the reviewers felt our paper provides a “Major theoretical result” [Reviewer 8SGa], that is also
 “novel” [Reviewer 13Yx, MsQo], and experimentally “comprehensive” [Reviewer MsQo] to an “important” [Reviewer 13Yx], “significant in the literature” [Reviewer DjaM] problem.

The primary concerns raised by the reviewers were regarding clarity in the proofs of concavity of our objective, and relation to the literature on finite dimensional convex-concave minimax optimization. We believe we have addressed these points, and summarize this below with more detailed responses given individually to the reviewers:

1) *(Reviewer DjaM) Appendix A.2 does not prove concavity to the mixture weights but concavity to the distribution* The objective being concave to the distribution implies the objective is concave to the mixture weights. We have clarified this implication in the proofs.


2. *(Reviewer MsQo) The relevance of the methodology to the existing literature on (empirical) group DRO is unclear.* The mentioned literature considers convex-concave minimax optimization over $\mathbb{R}^n$, whereas we look at optimization over bounded functions. Some key differences between these settings are: 1) we generally cannot assume a tractable gradient oracle over all bounded functions (e.g., the literature on gradient boosting), and so cannot directly use gradient based methods over our whole optimization space to find saddle points (unlike the mentioned work)  2) our problem is typically non-convex in the parameters so past convex methods no longer converge 3) our evaluation is on test group DRO performance (i.e., an estimate of the population DRO) so improvements to empirical group DRO (i.e., training loss) do not immediately translate, and in fact one often wants to restrict the optimization to generalize better when the model class is highly expressive as in our case.


3. *(Reviewer MsQo) This paper is missing comparisons to near sample optimal algorithms from the finite-dimensional convex literature.* After a close reading, we believe DoReMi in fact implements the near sample optimal Algorithm 1 of Zhang et al with additional hyperparameters. We now explicitly state this in the paper. Because we compared to DoReMi, we effectively have a comparison to Algorithm 1 of Zhang et al.  We thank the reviewer for bringing this up. Finally, one of our major applications is to non-parametric models, where the cited finite-dimensional algorithms do not apply.

---

### Author Response · Authors · 2024-12-03
**Summary of Discussion Period**

We thank the reviewers for their time during the discussion period, which led to meaningful improvements to our draft and several reviewers raising their score. To summarize the outcomes of the discussion period:

1. *Reviewer 13Yx:* We addressed their question about Bayes optimal functions, and they raised their score from a 6 to an 8.

2. *Reviewer DjaM:* We made changes to our draft which clarified the proofs of concavity, the main theorem, and also the role of the empirical gradient estimators. Given these clarifications the reviewer raised their score from a 5 to a 6.

3. *Reviewer MsQo:* We clarified how our method fits in the literature (adding citations), and the efficiency of our MixMax method alongside modifications to improve its efficiency. Given these changes the reviewer raised their score from a 3 to a 5.

We thank everyone for their time and effort. Our paper has truly become stronger from these discussions.

---

### Meta-Review · Area_Chair_F9ZP · 2024-12-19

**Metareview:**

The paper studies group DRO settings with non-convex losses and/or non-parametric model classes. The paper proposes reparameterizing the problem to function space and prove that group DRO over the space of bounded functions admits a minimax theorem. They further show that when the loss is squared or cross-entropy, the problem can be reformulated as a concave maximization problem. Finally, the paper provides a suite of experiments that demonstrate the effectiveness of the proposed approach. While there were initial concerns regarding clarity and novelty of the results, all were resolved in the rebuttal phase and the paper was judged well above the acceptance threshold.

**Additional Comments On Reviewer Discussion:**

The reviews raised several questions regarding novelty, relationship to prior work, and clarity of writing. The reviewers and authors engaged in a productive discussion, which led to multiple score increases. The authors have also appropriately revised the paper.

---

### Decision · Program_Chairs · 2025-01-22

Accept (Poster)